# Q-RAG: Long Context Multi-Step Retrieval via Value-Based Embedder Training

**Artyom Sorokin**[1,2,†], **Nazar Buzun**[1,3,*], **Alexander Anokhin**[2], **Egor Vedernikov**[2],
**Petr Anokhin**[1], **Mikhail Burtsev**[4], **Evgeny Burnaev**[1,2]

[1]AXXX, Moscow, Russia
[2]Applied AI Institute, Moscow, Russia
[3]Research Center of the Artificial Intelligence Institute, Innopolis University, Innopolis, Russia
[4]London Institute for Mathematical Sciences, London, UK
Correspondence to: griver29@gmail.com[†], n.buzun@seevia.ai[*]

## Abstract

Retrieval-Augmented Generation (RAG) methods enhance LLM performance by efficiently filtering relevant context for LLMs, reducing hallucinations and inference cost. However, most existing RAG methods focus on single-step retrieval, which is often insufficient for answering complex questions that require multi-step search. Recently, multi-step retrieval approaches have emerged, typically involving the fine-tuning of small LLMs to perform multi-step retrieval. This type of fine-tuning is highly resource-intensive and does not enable the use of larger LLMs. In this work, we propose Q-RAG, a novel approach that fine-tunes the Embedder model for multi-step retrieval using reinforcement learning (RL). Q-RAG offers a competitive, resource-efficient alternative to existing multi-step retrieval methods for open-domain question answering and achieves state-of-the-art results on the popular long-context benchmarks BabiLong and RULER for contexts up to 10M tokens. Code is available at: https://github.com/griver/Q-RAG.

## 1 Introduction

Large language models (LLMs) have achieved impressive results across a wide range of tasks (Novikov et al., 2025; Guo et al., 2025; Yang et al., 2025). However, they still face several fundamental limitations such as static knowledge, computational inefficiency on long contexts, degraded performance caused by attention dilution, and hallucinations (Hsieh et al., 2024; Kuratov et al., 2024; Liu et al., 2025). Retrieval-Augmented Generation (RAG) is one of the most widely used techniques to address these issues (Yu et al., 2024).

RAG works by extracting only the most relevant parts from a large external corpus or context, such as newly added knowledge or lengthy texts. This allows LLMs to operate on shorter and more focused inputs, improving efficiency and output quality. Most current RAG methods rely on single-step retrieval. This setup performs well in relatively simple tasks like Needle-in-a-Haystack (Hsieh et al., 2024). Still, more complex problems require multi-step interaction with the context. Multi-step retrieval can be viewed as a form of search-based reasoning. There are several existing approaches to multi-step retrieval reasoning. One direction involves constructing a knowledge graph from the retrieved information (Ma et al., 2025; Li et al., 2024). These methods are often slow at inference time, since the LLM must process the entire context to build the graph for each new input. Another line of work uses LLM agents, which interleave RAG queries with LLM-generated instructions (Singh et al., 2025; Anokhin et al., 2024). These systems are sensitive to noisy or inaccurate retrieved passages, which may disrupt the generation of future queries. This shows the need for joint optimization of the retrieval and generation components. Recently, methods have emerged that fine-tune LLMs to interact more effectively with retrieval tools (Song et al., 2025; Jin et al., 2025; Chen et al., 2025). These methods tend to perform better, but they require expensive fine-tuning of the LLM itself. This makes them impractical for large models and limits accessibility for most researchers and practitioners.

In this work, we focus on developing a resource-efficient multi-step RAG approach using reinforcement learning. Instead of fine-tuning an LLM, we train an agent that performs retrieval directly in the latent space of text chunk embeddings. This allows us to learn a compact and efficient model using value-based RL methods.

Our approach achieves state-of-the-art results on long-context commonsense reasoning, multi-hop QA, and NIAH tasks with contexts up to 10 million tokens. It also performs competitively on open-domain QA benchmarks such as MuSiQue and HotPotQA (Yang et al., 2018; Trivedi et al., 2022), while being significantly faster and cheaper to train and run compared to existing multi-step RAG methods. Our contributions are the following:

- We propose a new method for training a multi-step retrieval agent using temporal difference reinforcement learning.
- We achieve state-of-the-art results on benchmarks that require commonsense reasoning and NIAH tasks over ultra-long contexts (up to 10M tokens).
- We introduce a new way to incorporate temporal information into the multi-step embedder, enabling temporal reasoning during retrieval. Our temporal reasoning mechanism generalizes well to long contexts at inference time.

## 2 RELATED WORK

There are several main directions for tackling complex retrieval scenarios on long-context tasks.

A highly popular approach involves building fine-tuning-free LLM Agents that combine off-the-shelf retrievers with LLMs, such as Search-o1 (Li et al., 2025). Many of these works further enhance retrieval quality by constructing large knowledge graphs over the context, which, while requiring little additional training, are extremely slow at inference due to the need for LLMs to process the entire context, e.g. GraphReader (Li et al., 2024), HippoRAG (Jimenez Gutierrez et al., 2024), AriGraph (Anokhin et al., 2024).

Another line of work fine-tunes LRMs to perform multi-step retrieval, allowing the model to generate intermediate search queries inside the reasoning for long contexts. The first work to apply this idea was IM-RAG (Yang et al., 2024), which fine-tuned the LLM with a frozen embedder using PPO (Schulman et al., 2017). More recent papers, such as R1-Searcher (Song et al., 2025), Search-R1 (Jin et al., 2025), RAG-RL (Huang et al., 2025), and ReSearcher (Chen et al., 2025), extended this direction by employing GRPO (Shao et al., 2024) for the task. Unlike these methods, which freeze the embedder and fine-tune the LLM, our approach fine-tunes only the embedder, allowing it to pair with LLMs of any size, including proprietary ones, while keeping fine-tuning efficient and inexpensive.

A different approach is to fine-tune the retriever itself using feedback from the LLM, as in RePlug (Shi et al., 2024). This direction is most similar to ours, but RePlug did not address multi-step reasoning or use reinforcement learning in this setting. BeamRetriever (Zhang et al., 2024) achieves state-of-the-art results on short-context QA by training a reranker for BeamSearch-style planning. In contrast, Q-RAG trains the embedder with reinforcement learning, enabling faster inference and better scalability to long contexts through efficient vector similarity instead of transformer-based trajectory scoring.

Extremely long-sequence processing is demonstrated by models that combine recurrence with the Transformer architecture. The Mamba family of state space models (Gu & Dao, 2024) replaces attention with structured recurrent dynamics, offering linear-time scalability and strong performance on long sequences, though often at the cost of weaker in-context learning and less expressive token-to-token interaction compared to Transformer-based architectures. The Recurrent Memory Transformer (RMT) (Bulatov et al., 2022) introduces segment-level recurrence by passing memory tokens between fixed-size segments, enabling Q&A on sequences up to 10M tokens. Titans (Behrouz et al., 2024) frames recurrent memory training as a meta-learning problem and uses surprise to prioritize information that should be retained over very long contexts, showing scaling beyond 2M tokens. Relatedly, MemUP (Sorokin et al., 2022) used uncertainty to identify events that require long-term memory in recurrent models. Similar to Titans, ATLAS (Behrouz et al., 2025) increases memory capacity, achieving better long-context performance than both RMT and Titans. The Associative

Recurrent Memory Transformer (ARMT) (Rodkin et al., 2024) employs quasi-linear, associative attention in each layer and attains the best long-context scores among recurrent models. Our approach outperforms all of these models on contexts beyond 1M tokens while belonging to a different class of methods.

LongRoPE2 (Shang et al., 2025) tackles the positional encoding bottleneck, extending the effective context window of pre-trained LLMs to 128K tokens while retaining short-context performance through RoPE rescaling and mixed-window training.

# 3 METHODS

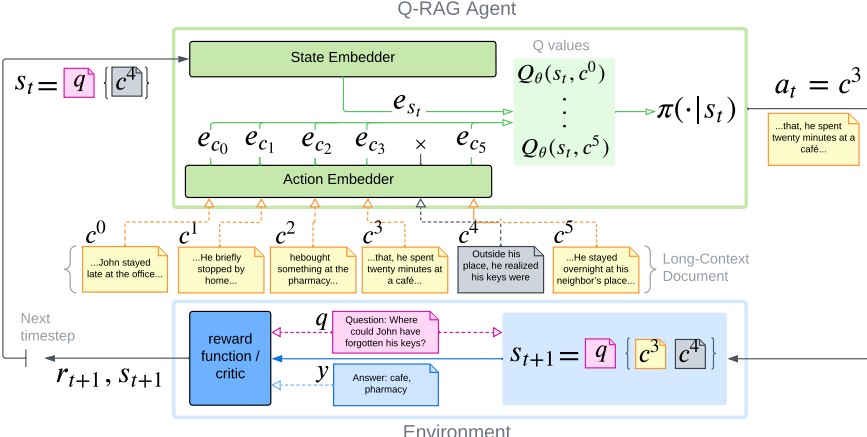

Figure 1: Q-RAG agent interacts with multi-step retrieval environment. The starting state $s_0$ contains the initial query $q$. At the start of the episode, the agent embeds all chunks of the long context $\mathbb{C}$. At each step $t$, the agent computes a vector embedding of the current state $s_t$, which includes $q$ and all previously selected chunks. For every chunk $c^i \in \mathbb{A}_t$, the utility of retrieving it is evaluated by the $Q$-function $Q_\theta(s_t, a = c^i)$. The policy $\pi_\theta$ selects the next chunk from $\mathbb{A}_t$ with probability proportional to its $Q_\theta(s_t, c^i)$ value.

## 3.1 PRELIMINARIES

Let $\mathcal{D}$ be a dataset of triples $(\mathbb{C}, q, y)$, where $\mathbb{C}$ is a long context, $q$ is an initial query, and $y$ is the gold answer. The query $q$ can be either a user question about $\mathbb{C}$ or a generated claim whose factuality or consistency with earlier parts of $\mathbb{C}$ must be verified. We assume $\mathbb{C}$ is pre-segmented into non-overlapping[1] text chunks $\mathbb{C} = \{c^{(i)}\}_{i=1}^m$ in document order. The agent's goal is to identify the information in $\mathbb{C}$ that is missing from $q$ but necessary to produce the correct answer $y$. We model multi-step retrieval as a finite-horizon Markov Decision Process, or MDP $(\mathbb{S}, \mathbb{A}, p, r, \gamma)$, where $\mathbb{A}$ is the action space, $\mathbb{S}$ is the state space, $r$ is the reward function, $p$ is the (deterministic) transition function, and $\gamma \in [0, 1]$ is the discount factor. At step $t = 0$, the action set is $\mathbb{A}_0 = \mathbb{C}$, where an action $a_t \in \mathbb{A}_t$ selects one chunk. At later steps, previously selected chunks are removed so $\mathbb{A}_t = \mathbb{C} \setminus \{a_0, \ldots, a_{t-1}\}$. Superscripts indicate document positions and subscripts indicate episode timesteps. The notation $a^i$ (equivalently $c^{(i)}$) denotes the chunk/action at position $i$ in the document; selecting the chunk with index $i$ at step $t$ is written $a_t^i$. Symbols $c$ and $a$ are used interchangeably, depending on context.

States are ordered lists that always begin with the query, $s_t = \mathrm{ord}([q, a_0, \ldots, a_{t-1}])$, where $\mathrm{ord}(\cdot)$ sorts by the original document order to avoid permutation ambiguity; the initial state contains only the query, $s_0 = [q]$. Transitions are deterministic, $p(s_t, a_t) = \mathrm{ord}([q, a_0, \ldots, a_{t-1}, a_t])$. An episode terminates either when a step budget $T$ is reached or when a special STOP action is taken.

---

[1] Chunk overlapping may complicate the explanation but does not affect our proposed solution.

When supervision provides a set of support facts $F^\star \subseteq C$, we use a sparse terminal reward: the reward is 0 at all intermediate steps, and at the end of the episode it is 1 if all support facts are included in the final state (otherwise 0). When only answer supervision is available, one could instead use an LLM to generate $\hat{y}$ from the final state and define a terminal reward via an answer-quality metric (e.g., exact match or F1). In this work we do not pursue LLM-based rewards; all reported experiments rely on the support-fact signal, and exploring LLM-based reward design is left for future work.

## 3.2 Value-based RL for Embedder Fine-Tuning

Action selection in multi-step retrieval is performed by a value-based agent. Specifically, maximum-entropy reinforcement learning (Ziebart, 2010; Haarnoja et al., 2018) is adopted together with the corresponding definitions of the soft $Q^\pi$ and $V^\pi$ value functions for a policy $\pi$:

$$Q^\pi(s,a) = r(s,a) + \gamma V^\pi(s' = p(s,a)) \tag{1}$$

$$V^\pi(s) = \mathbb{E}_{a \sim \pi(\cdot|s)} \left[ Q^\pi(s,a) - \alpha \log \pi(a|s) \right] \tag{2}$$

Here, $\alpha > 0$ is a temperature that controls the strength of exploration. This choice is primarily motivated by the need for effective exploration in the long-context multi-step retrieval environment. In Q-RAG, the Q-function is approximated using two embedders for states and actions. The state embedder $E_s(s_t; \theta_1) \in \mathbb{R}^d$ produces a vector embedding for the current state $s_t$, while the action embedder $E_a(a^i, i; \theta_2) \in \mathbb{R}^d$ employs rotary position embeddings to encode both the candidate chunk content and its document-position index $i$. Q values are then estimated by an inner product between two embeddings: $Q_\theta(s, a^i) = \langle E_s(s; \theta_1), E_a(a^i, i; \theta_2) \rangle$. This factorization is theoretically grounded; we derive its convergence guarantees with explicit rates in Appendix A. Given $Q_\theta$, the chunk selection probability is computed using a Boltzmann policy:

$$\pi(a_t|s_t) = \frac{\exp \frac{1}{\alpha} (Q_\theta(s_t, a_t) - q)}{\sum_{a \in \mathcal{A}_t} \exp \frac{1}{\alpha} (Q_\theta(s_t, a) - q)} \tag{3}$$

with $q = \max_{a \in \mathcal{A}_t} Q_\theta(s_t, a)$ and temperature $\alpha$ annealed from an initial value to zero during training (proportionally to the learning rate).

As the backbone Temporal Difference learning algorithm, we adopt the recent PQN method by Gallici et al.. Compared to DQN (Mnih et al., 2015), PQN removes the need for a replay buffer. In our setting with a large number of chunks, a replay buffer would require re-embedding all document chunks for each sample drawn from the replay buffer to estimate $V/Q$ values for subsequent states $s_{t+1}$. This significantly slows the training process and increases memory requirements. Using PQN enables an on-policy value-based training that avoids these costs. The key departures in Q-RAG, relative to the original PQN backbone, are the use of soft value functions and target networks. Ablation results demonstrating the benefit of these choices are reported in Section 5.

As the training target, rather than the one-step return (see r.h.s. in Eq. 1), a $\lambda$-return is used to improve stability and learning speed:

$$G_t^\lambda = (1 - \lambda) \sum_{n=1}^{T-t-1} \lambda^{n-1} G_{t:t+n} + \lambda^{T-t-1} G_t,$$

where $G_{t:t+n} = \sum_{k=1}^n \gamma^{k-1} r_{t+k} + V_{\theta'}(s_{t+n})$. The approximation of the state value function can be computed from Q values in the case of discrete actions:

$$V_{\theta'}(s_t) = \alpha \log \sum_{a \in \mathcal{A}_t} \exp \left( \frac{Q_{\theta'}(s_t, a)}{\alpha} \right) \tag{4}$$

Here $\theta'$ denotes slowly updated target network parameters. The model parameters $\theta$ are fine-tuned to minimize the mean squared error to the $\lambda$-returns:

$$\mathcal{L}_Q = \mathbb{E}[(Q_\theta(s_t, a_t) - G_t^\lambda)^2] \tag{5}$$

The Q-RAG pseudocode is presented in Algorithm 1.

---

**Algorithm 1** Q-RAG

---

1: **Hyperparameters:**
2:     Number of environments $K$, retrieval steps $T$, temperature $\alpha$, TD parameter $\lambda$, EMA $\tau$.
3: **Initialize:**
4:     State embedder $E_s(s; \theta_1)$
5:     Action embedder $E_a(a^i, i; \theta_2)$ with position $i$
6:     Critic $Q_\theta(s, a^i) = E_s(s; \theta_1)^T E_a(a^i, i; \theta_2)$
7:     Critic target $Q_{\theta'}(s, a^i)$
8: **procedure** COMPUTETARGETS($\{s_t, a_t, r_t, v_t\}_{t=1}^{T+1}$)
9:     Initialize $\lambda$-returns $G_T = r_T + \gamma v_{T+1}$
10:     **for** $t = T - 1$ **downto** 1 **do**
11:         $G_t = r_t + \gamma\big[(1 - \lambda)v_{t+1} + \lambda G_{t+1}\big]$
12:     **end for**
13:     **return** $\{G_t\}_{t=1}^T$
14: **end procedure**
15: Training (one update step)
16: **for** env $k \in 1, \ldots, K$ **in parallel do**
17:     $s_1, \mathcal{A}_1 = $ ResetQueryAndContext()
18:     Compute $E_a = E_a(\mathcal{A}; \theta)$ and $E_a' = E_a(\mathcal{A}; \theta')$
19:     **for** step $t \in 1, \ldots, T + 1$ **do**
20:         $a_t \sim \text{softmax}_{a \in \mathcal{A}_t} \frac{1}{\alpha} E_s(s; \theta)^T E_a$
21:         $v_t = \alpha \log \sum_{a \in \mathcal{A}} \exp \frac{1}{\alpha} E_s(s; \theta')^T E_a'$
22:         $r_t = \text{ComputeReward}(s_t, a_t)$
23:         $s_{t+1} = \text{concatenate}(s_t, a_t)$
24:         $\mathcal{A}_{t+1} = \mathcal{A}_t \setminus \{a_t\}$
25:     **end for**
26:     $\mathcal{B} = \{s_t, a_t, r_t, v_t\}_{t=1}^{T+1}$
27:     $\{G_t^k\}_{t=1}^T = \text{ComputeTargets}(\mathcal{B})$
28: **end for**
29: $\nabla \mathcal{L}_Q = \frac{1}{TK} \sum_{k=1}^K \sum_{t=1}^T \nabla_\theta (Q_\theta(s_t^k, a_t^k) - G_t^k)^2$
30: Update $\theta$ using $\nabla \mathcal{L}_Q$
31: Update target parameters: $\theta' \leftarrow \tau\theta + (1 - \tau)\theta'$

---

### 3.3 TEMPORAL REASONING FOR LONG-CONTEXT SEARCH

When dealing with narrative text, the information contained in a text chunk $c$ may be insufficient to determine whether $c$ helps us answer the question $q$. For example, we may need to know what happened before some specific event. A standard retriever can find several relevant text chunks that specify the character's location, but choosing the correct one can be impossible without taking into account temporal information. To address this, we propose a *relative positional encoding* of chunks that explicitly encodes their position with respect to the facts already extracted into the state. At step $t$, let $S_t = \{i_1 < \cdots < i_k\}$ be the (sorted) document indices of selected chunks and $\mathbb{A}_t$ the set of available actions. The indices in $S_t$ partition the document into $k+1$ disjoint intervals: "before the earliest selected fact", "between consecutive selected facts", and "after the latest selected fact." The relative positional mapping $\rho_t : \mathbb{N} \to \mathbb{R}^+$ assigns to every original chunk index a real-valued index that (i) identifies the interval it belongs to and (ii) preserves the relative order between chunks. This mapping makes explicit *between which extracted facts* a chunk lies, while remaining invariant to global shifts of absolute positions.

Formally, the interval boundaries are defined as $b_0=1$, $b_j=i_j$ for $j=1{:}k$, and $b_{k+1}=m+1$ for $\mathbb{C} = \{c^{(i)}\}_{i=1}^m$. To compute relative index $\rho_t(i)$ for a chunk $c^i$, find the unique $j$ such that $b_j \le i < b_{j+1}$ and set

$$\rho_t(i) \;=\; j\,\delta \;+\; \ell\,\frac{i - b_j}{b_{j+1} - b_j}, \tag{6}$$

where $\delta > 0$ is the inter-interval step and $\ell \in (0, \delta)$ controls the within-interval resolution (e.g., $\delta=10$, $\ell=9$ in our experiments). In the action embedder, the absolute position is replaced by the

relative one,

$$E_a(a^i, i; \theta_2) \Rightarrow E_a(a^i, \rho_t(i); \theta_2), \tag{7}$$

which allows the Q-function to exploit the spatial relation of candidates to already retrieved evidence while retaining local order within each interval. This design allows the retrieval agent to perform strongly not only on fact-finding over disjoint document collections, but also on long-form narrative tasks, enabling Q-RAG to compete with recurrent transformers (Bulatov et al., 2022; Rodkin et al., 2024; Behrouz et al., 2025; 2024) and other long-context approaches.

## 4 EXPERIMENTS

### 4.1 EXPERIMENTAL SETUP

We evaluate our approach, Q-RAG, on tasks that cover commonsense reasoning, temporal reasoning, a set of Needle-in-a-Haystack tasks and open-domain multi-hop question answering tasks on context lengths that range from 4k tokens to 10M tokens per sample. For commonsence and temporal reasoning we use **BabiLong** benchmark (Kuratov et al., 2024), for Needle-in-a-Haystack, we use the **RULER** benchmark (Hsieh et al., 2024). For open-domain multi-hop QA we use **HotPotQA** (Yang et al., 2018), **MuSiQue** (Trivedi et al., 2022) and **RULER** benchmarks. BabiLong and RULER require long contexts. MuSiQue and HotPotQA use short contexts.

Baselines differ by task. Computing a uniform set of baselines across all datasets is difficult and time-consuming. Many methods do not release code. Some methods were evaluated only on some of these datasets. Even when the tasks match, the experimental settings often differ for the same benchmarks. Some baselines provide code but require heavy resources, for example at least 8×A100 GPUs (Jin et al., 2025; Song et al., 2025; Huang et al., 2025)) to fine-tune, which are unavailable to us. Therefore, we report three types of baselines, and we mark each baseline in tables accordingly:

- $^\times$ **Ablation**: baselines that test the effectiveness of our proposed modifications.
- $^\checkmark$ **Reproduced**: baselines that we fine-tuned and/or evaluated on our datasets using released code or publicly available checkpoints.
- $^\circ$ **Reported**: baselines whose scores we take directly from the original papers.

### 4.2 COMMONSENSE REASONING ON ULTRA-LONG CONTEXTS

On the BabiLong (Kuratov et al., 2024) benchmark, we compared our method with the state-of-the-art long-context processing approaches, including Titans (Behrouz et al., 2024), Atlas (Behrouz et al., 2025), ARMT (Rodkin et al., 2024), RMT (Bulatov et al., 2022), as well as proprietary LLMs and LLM-based agents. The results for most of these baselines were taken directly from the respective original papers. As shown in Figure 2a, our approach achieves the highest average

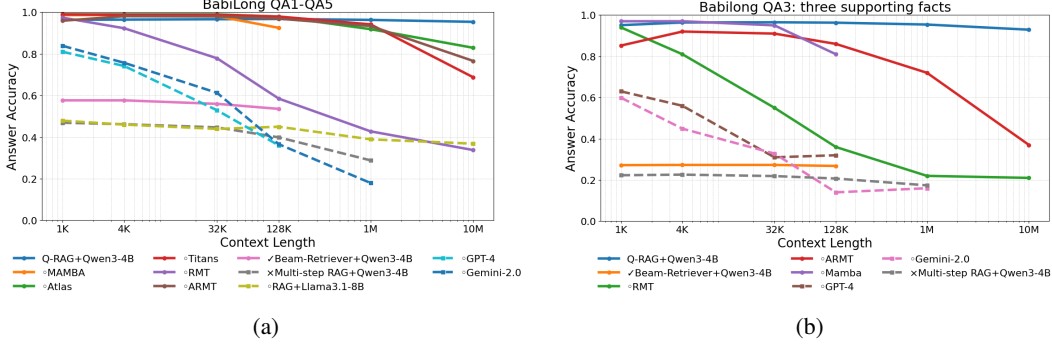

Figure 2: Comparison of answer accuracy on the long-context benchmark BabiLong. Solid lines denote methods fine-tuned on BabiLong, while dashed lines denote zero-shot methods. **a)** Average performance across tasks Q1–QA5. **b)** Performance on the hardest task, QA3, which requires the longest reasoning chain and temporal awareness.

performance on BabiLong in ultra-long contexts ranging from 1 to 10 million tokens, demonstrating superior generalization to long contexts compared to other specialized long-context methods.

In Figure 2b, we present separate results for the QA3 subtask, which is the hardest subtask in the BabiLong benchmark, it requires multi-step search of at least three different facts and temporal reasoning. Experimental results show that the majority of models perform worst on the QA3 subtask. As the results indicate, alternative long-context approaches show even greater performance degradation on this task with increasing context length. In contrast, Q-RAG shows virtually no degradation, with the largest performance gap over all baselines observed on this most challenging subtask. We additionally fine-tuned the Beam Retriever baseline specifically on the QA3 subtask, given its strong performance on open-domain QA datasets. However, this method failed to solve the task. Note that some methods, such as Titans (Behrouz et al., 2024) and Atlas (Behrouz et al., 2025), are absent from the figure as they did not report detailed breakdowns by subtask.

## 4.3 NEEDLE-IN-A-HAYSTACK AND LONG CONTEXT QA

While reasoning tasks are crucial for evaluating advanced retrieval systems, a substantial portion of real-world applications reduces to Needle-in-a-Haystack (NIAH) problems, making it equally important that models deliver consistently strong performance on these tasks. RULER is a dataset that includes many long-context tasks. Most of these tasks follow the NIAH formulation. The NIAH setup evaluates the ability to retrieve a specific "needle" from a long distracting "haystack". For the RULER benchmark, we use Beam Retriever (Zhang et al., 2024), Titans (Behrouz et al., 2024), Atlas (Behrouz et al., 2025), Mamba2 (Waleffe et al., 2024), and LongRoPE2 (Shang et al., 2025) as baselines. Titans and Atlas are recurrent transformers. Mamba2 is a state space model (SSM) that combines transformer components with SSM. LongRoPE2 is a method for extending the

Table 1: Results on the RULER benchmark, evaluating long-context retrieval performance across various context lengths. **S** (Single-needle): Find one value for one key. **MK** (Multi-keys): Find one value for one key among many. **MV** (Multi-values): Find all values for one key. **MQ** (Multi-query): Answer multiple questions over the context. **MH QA**: open-domain multi-hop question answering. **SH QA**: single-hop question answering.

| Len | Methods | S | | | MK | | | MV | MQ | NIAH Avg. | QA | |
|---|---|---|---|---|---|---|---|---|---|---|---|---|
| | | 1st | 2nd | 3rd | 1st | 2nd | 3rd | | | | SH | MH |
| 4K | °Titans | 98.4 | 99.8 | 89.4 | n/a | n/a | n/a | n/a | n/a | n/a | n/a | n/a |
| | °Atlas | 99.2 | 100 | 90.6 | n/a | n/a | n/a | n/a | n/a | n/a | n/a | n/a |
| | °Mamba2-Hybrid | 100 | 100 | 95.7 | 89.5 | 95.5 | 96 | 97.9 | 97.6 | 96.5 | 56.5 | 48.8 |
| | °LongRoPE2-8B | 100 | 100 | 99 | 100 | 100 | 100 | 99 | 99.7 | 99.7 | 79 | 60 |
| | ✓Beam Retriever | 100 | 100 | 98 | 98 | 98 | 97 | 98 | 99 | 98.5 | 29.0 | 39.0 |
| | Q-RAG | 100 | 100 | **100** | 100 | 100 | 100 | **100** | **100** | **100** | 62 | 67 |
| 16K | °Titans | 96.2 | 80.2 | n/a | n/a | n/a | n/a | n/a | n/a | n/a | n/a | n/a |
| | °Atlas | 97 | 84 | n/a | n/a | n/a | n/a | n/a | n/a | n/a | n/a | n/a |
| | °Mamba2-Hybrid | 100 | 100 | 81.5 | 92 | 92.2 | 83 | 89.8 | 90.2 | 91.1 | 48.8 | 44 |
| | °LongRoPE2-8B | 100 | 100 | 100 | 99 | 100 | 98 | 95 | 98.2 | 98.8 | 69 | 58 |
| | ✓Beam Retriever | 100 | 100 | 97 | 96.5 | 96 | 95 | 80 | 98 | 95.3 | 24.0 | 35.0 |
| | Q-RAG | 100 | 100 | 100 | **100** | 100 | **100** | **100** | **100** | **100** | 59 | 64 |
| 32K | °Mamba2-Hybrid | 100 | 100 | 96.7 | 84 | 76.5 | 81.5 | 84.3 | 80.9 | 88.0 | 41.8 | 38.5 |
| | °LongRoPE2-8B | 100 | 100 | 100 | 99 | 98 | 100 | 98 | 96.2 | 98.9 | **72** | 55 |
| | Q-RAG | 100 | 100 | 100 | **100** | **100** | 100 | **100** | **100** | **100** | 59 | 65 |
| 128K | °LongRoPE2-8B | 100 | 100 | 99 | 96 | 91 | 94 | 96.5 | 97 | 96.7 | **56** | 50 |
| | Q-RAG | 100 | 100 | **100** | **100** | **100** | **100** | **100** | **100** | **100** | 55 | 65 |
| 1M | Q-RAG | 100 | 100 | 100 | 100 | 98.5 | 99.0 | 100 | 100 | 99.7 | 52 | 61 |

effective context window of LLMs. All methods were fine-tuned either directly on RULER (Titans, Atlas, Mamba2, Beam Retriever) or on related synthetic NIAH-style datasets (LongRoPE2). Q-RAG was also fine-tuned on the NIAH subtasks. For the Multi-hop QA RULER subtask, Q-RAG and Beam Retriever were fine-tuned on HotPotQA and evaluated on the Multi-hop QA subtask out-of-distribution.

The results are shown in Table 1. Q-RAG achieves near-perfect performance on all NIAH subtasks. The Q-RAG embedder was trained on 4K-length documents and generalizes to context lengths up to 1M tokens without loss of accuracy. On the Multi-hop QA subtask, Q-RAG shows significantly better results than all our baselines at all context lengths we consider. Some degradation with increasing context length begins only at 1M tokens.

## 4.4 Open-domain Question Answering

For our experiments on the HotPotQA and MuSiQue datasets, we compared our method against several strong baselines. The first baseline is Beam Retriever, which enables multi-step retrieval by training a model to score sequences of retrieved chunks. During evaluation, Beam Retriever is given the oracle number of supporting facts (i.e., the gold hop count) and always retrieves exactly that many facts. Although this approach is slower than traditional retrieval methods and does not scale well to longer contexts, it achieves state-of-the-art results on HotPotQA. Another baseline we considered is SearchR1, a recent method from a family of approaches that train the LLM itself to compose text queries for multi-step retrieval. Additionally, we evaluated the performance of LLM-agent-based methods, including GraphReader. Q-RAG and Beam Retriever were fine-tuned on HotPotQA and evaluated on MuSiQue for out-of-distribution testing. Baseline numbers were taken directly from the corresponding papers. Missing entries indicate metrics not reported by the original authors.

The comparison results are presented in Table 2. Our method achieves fact retrieval accuracy on par with Beam Retriever, surpasses all other baselines on HotPotQA, and matches the performance of full-LLM-tuning Search-R1 while outperforming all alternatives on the out-of-distribution MuSiQue dataset, resulting in the best overall performance across benchmarks. Results also include another Q-RAG version *Plan Q-RAG* that combines the Q-RAG value function and beam search based planning (see Appendix C). Plan Q-RAG showed similar performance to vanilla Q-RAG. For both methods involving retrieval mechanism fine-tuning (Q-RAG and Beam Retriever), we used the QwQ-32B model to produce the final answer.

Table 2: Comparison of methods on HotPotQA and MuSiQue benchmarks. Bold text and underline denote the best and second best scores respectively.

| Methods | HotPotQA | | | | MuSiQue (OOD) | | | | Avg | |
|---|---|---|---|---|---|---|---|---|---|---|
| | Fact F1 | Fact EM | Ans F1 | Ans EM | Fact F1 | Fact EM | Ans F1 | Ans EM | Ans F1 | Ans EM |
| Fine-tuned on HotPotQA | | | | | | | | | | |
| Plan Q-RAG | 0.95 | 0.91 | 0.76 | 0.60 | 0.69 | 0.53 | 0.51 | 0.36 | **0.64** | **0.48** |
| Q-RAG | 0.93 | 0.89 | 0.76 | 0.59 | **0.71** | **0.55** | **0.52** | 0.37 | **0.64** | **0.48** |
| ✓Beam Retriever | **0.97** | **0.94** | **0.77** | **0.61** | 0.61 | 0.36 | 0.40 | 0.27 | 0.59 | 0.44 |
| ✓Search-r1 | 0.81 | 0.66 | 0.65 | 0.52 | **0.71** | **0.55** | 0.51 | **0.39** | 0.58 | 0.46 |
| °RAG-RL | 0.82 | – | 0.69 | – | 0.65 | – | 0.47 | – | 0.58 | – |
| ×Multi-step RAG w.o. FT | 0.73 | 0.54 | 0.65 | 0.50 | 0.51 | 0.30 | 0.40 | 0.27 | 0.53 | 0.39 |
| Zero-shot methods | | | | | | | | | | |
| ✓GraphReader | – | – | 0.46 | 0.24 | – | – | 0.40 | 0.20 | 0.43 | 0.22 |
| ✓Single-step RAG | – | – | 0.53 | 0.39 | – | – | 0.28 | 0.17 | 0.41 | 0.28 |

## 5 Ablation Study

To assess the impact of the architectural choices in Q-RAG, an ablation study was conducted on the BabiLong-QA3 task. This benchmark was selected because it is among the most challenging

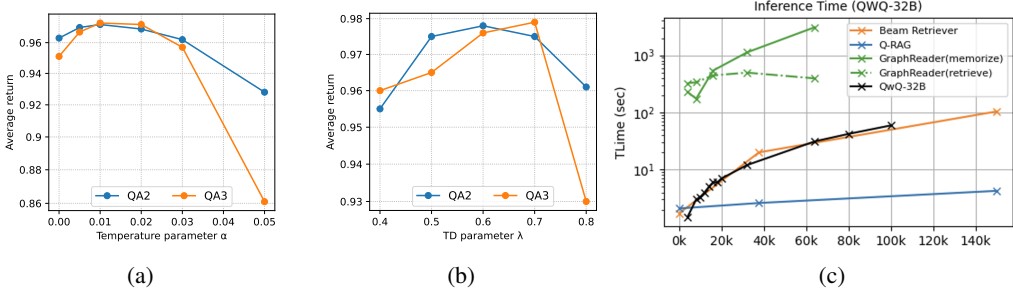

Figure 3: Ablation for (a) policy entropy coefficient ($\alpha$) in soft Q function and (b) for $\lambda$-return parameter. Inference runtime comparison (c), context length in tokens on the x-axes.

long-context tasks used in the experiments and it supports evaluation at arbitrary context lengths. The following baselines were compared against Q-RAG:

**Multi-step RAG w.o. FT.** This baseline reproduces the full Q-RAG retrieval pipeline and uses the same state and action embedders, but relies on their original pretrained weights without any reinforcement learning fine-tuning. This setting tests whether RL fine-tuning of the embedders is beneficial for multi-step retrieval quality.

**Multi-step RAG w. SFT.** This baseline applies supervised fine-tuning using ground-truth support facts as supervision. The loss follows the objective used in BeamRetriever for trajectory supervision, adapted to the multi-step retrieval setting. This setting isolates the effect of RL by comparing it to supervised learning on the same supervision signal.

**Q-RAG w.o. target.** This variant removes target networks from the PQN-based value learning, following the original PQN recipe without target parameters. It measures the contribution of target networks to stability and performance in the Q-RAG training loop.

**Q-RAG w.o. Soft-Q.** This variant replaces the maximum-entropy (soft) value functions with standard (non-entropy-regularized) Q-learning objectives. It evaluates the effect of entropy regularization and the soft value formulation on retrieval performance.

All baselines were evaluated with three random seeds. Table 3 reports results across multiple context lengths on QA3. Figure 3 shows the sensitivity of Q-RAG to the $\lambda$-return parameter and the temperature $\alpha$ (the strength of entropy regularization) on QA2 and QA3.

Table 3: Ablation results on BabiLong QA3. The Table shows F1 score for supporting facts retrieval. All values are averaged over 3 runs with different seeds.

| Method | 1K | 4K | 32K | 128K | 1M |
|---|---|---|---|---|---|
| Q-RAG | $97.8 \pm 0.17$ | $97.4 \pm 0.14$ | $97.1 \pm 0.08$ | $96.8 \pm 0.08$ | $96.5 \pm 0.16$ |
| [×]Q-RAG w.o. Soft-Q | $95.9 \pm 0.70$ | $95.5 \pm 0.80$ | $94.5 \pm 0.50$ | $94.0 \pm 0.30$ | $93.3 \pm 0.45$ |
| [×]Q-RAG w.o. Target | $79.2 \pm 26.0$ | $78.1 \pm 26.6$ | $77.6 \pm 27.2$ | $77.4 \pm 27.3$ | $75.9 \pm 28.2$ |
| [×]Multi-Step RAG w. SFT | $20.33 \pm 0.32$ | $20.87 \pm 0.35$ | $20.10 \pm 0.20$ | $18.30 \pm 0.36$ | — |
| [×]Multi-Step RAG w.o. FT | $15.52 \pm 0.11$ | $16.38 \pm 0.10$ | $15.51 \pm 0.16$ | $15.34 \pm 0.12$ | — |

## 5.1 SENSITIVITY TO RETRIEVAL BUDGET

We investigate the dependence of final model performance on the number of Q-RAG retrieval steps (i.e., the retrieval budget). For this analysis, we used a Q-RAG system with an Alibaba-NLP/gte-multilingual-base embedder, trained on a combination of the HotPotQA and MuSiQue datasets. This embedder supports contexts of up to 8192 tokens, enabling the use of a larger retrieval budget. We evaluated the system on 1000 samples from the HotPotQA dataset. The final generation of the answers was performed by three LLMs: Qwen3-4B, Qwen3-14B, and Qwen3-32B.

Table 4: Sensitivity to the number of retrieval steps. Dataset: HotPotQA (1000 samples). Embedder Alibaba-NLP/gte-multilingual-base was trained on HotPotQA+MuSiQue.

| Retrievals | Facts | | Qwen3-4B | | Qwen3-14B | | Qwen3-32B | |
|---|---|---|---|---|---|---|---|---|
| | EM | F1 | EM | F1 | EM | F1 | EM | F1 |
| 2 | 0.832 | 0.903 | 0.439 | 0.620 | 0.556 | 0.708 | 0.504 | 0.675 |
| 3 | 0.935 | 0.771 | 0.481 | 0.657 | 0.570 | 0.730 | 0.510 | 0.692 |
| 4 | 0.962 | 0.652 | 0.493 | 0.664 | 0.577 | 0.734 | 0.513 | 0.695 |
| 5 | 0.978 | 0.565 | 0.481 | 0.656 | 0.584 | 0.744 | 0.512 | 0.692 |

The results are presented in Table4. Here, EM (Exact Match) indicates the number of correct (ground-truth supporting) chunks retrieved, while F1 accounts for the inclusion of noise (non-supporting) chunks. The table shows that increasing the number of retrieval steps from 2 to 3 improves both the number of correct facts retrieved and the answer quality across all three LLMs. These experiments suggest that, within a reasonable range of retrieval counts, final answer accuracy is primarily dependent on the retrieval of correct chunks and is not degraded by the presence of noise chunks.

In addition to the fixed-budget setting, we also studied an alternative stopping criterion in which the agent stops dynamically according to a Q-value threshold. A detailed analysis of this Q-value-based early stopping is presented in Appendix B.

## 6 CONCLUSION

This work introduced Q-RAG, a resource-efficient method for multi-step retrieval trained with reinforcement learning directly in the latent space of text-chunk embeddings. Across long-context benchmarks (e.g., *BabiLong*, *RULER*) and open-domain QA datasets (e.g., *MuSiQue*, *HotPotQA*), Q-RAG attains state-of-the-art or highly competitive results. Its advantage over baselines widens as context length grows, and performance shows minimal degradation even at ultra-long scales. A key practical benefit is compute efficiency: all training was performed on a single A100 GPU with 80 GB memory, whereas recent RL-based multi-step retrievers such as Search-R1/R1-Searcher typically report training on clusters of about eight A100 GPUs. By fine-tuning only the embedder while keeping the LLM frozen, Q-RAG remains easy to pair with powerful pre-trained or proprietary LLMs, enabling efficient training, flexible deployment, and strong retrieval over very long contexts. Looking ahead, promising directions include using structured LLM feedback as a reward signal, strengthening compositional and temporal reasoning directly in the embedding space, and exploring tighter integration with generation while preserving the method's efficiency and scalability.

REPRODUCIBILITY STATEMENT.

The main results of this paper can be reproduced using the code available in the GitHub repository, which includes data and instructions for running experiments on all benchmarks: BABILong, HotPotQA, MuSiQue, and RULER. Pretrained Q-RAG checkpoints are available in the Hugging Face repository. Only publicly available embedders are fine-tuned: `multilingual-e5-large`, `Alibaba-NLP/gte-multilingual-base`, and `facebook/contriever`. The hyperparameters and training schedules are provided in the code and in Appendix F. A single NVIDIA A100 Tensor Core GPU with 80 GB of memory is sufficient to reproduce all Q-RAG experiments.

ACKNOWLEDGMENTS

We thank Oleg Inozemcev for running additional experiments and assisting with hyperparameter tuning during the rebuttal stage, which helped improve several results. We also thank Alexey Trushkov and Wenshuai Yin for their valuable feedback and constructive criticism, which helped improve the paper. The work was partially supported by the grant for research centers in the field of AI provided by the Ministry of Economic Development of the Russian Federation in accordance with the agreement 000000C313925P4F0002 and the agreement №139-10-2025-033

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

## A    INNER PRODUCT APPROXIMATION FOR Q-FUNCTION

The classical Universal Approximation Theorem (UAT) asserts that sufficiently expressive neural networks can approximate any continuous function on a compact domain arbitrarily well. In this section, we prove a variant of the UAT for functions decomposed as an inner product between state embeddings and action embeddings modulated by a positional block-diagonal matrix.

Let $X \subset \mathbb{R}^{d_x}, Y \subset \mathbb{R}^{d_y}$ and $T \subset \mathbb{R}$ be compact sets and define $K = X \times Y \times T$. We will approximate any $f \in C(K, \mathbb{R})$ in the uniform norm. One may identify $X$ with the environment state space $\mathbb{S}$, $Y$ with the set of available actions $\mathbb{A}$ (see Section 3), and interpret $t \in T$ as a relative positional encoding for actions[2]. Under this correspondence, a function $f(x, y, t)$ represents a ground-truth $Q$-function. If the $Q$-function does not depend on position $t$, one can simply take $f$ to be constant in $t$.

Our starting point is the real-valued score function

$$F'(x, y, t) = \langle h_{\mathbb{R}}(x), R_t\, g_{\mathbb{R}}(y) \rangle_{\mathbb{R}^{2m}}, \tag{8}$$

where $m \in \mathbb{N}$ is arbitrary, $h_{\mathbb{R}} : X \to \mathbb{R}^{2m}$ and $g_{\mathbb{R}} : Y \to \mathbb{R}^{2m}$ are continuous encoders (e.g., neural networks), and $R_t \in \mathbb{R}^{2m \times 2m}$ is a position-dependent *block rotation* acting independently on each coordinate pair. The standard Rotary Position Embedding (RoPE) is precisely a family $t \mapsto R_t$ of this type.

A useful reformulation is that every real-valued score of the form equation 8 can be written as the real part of a complex inner product after identifying $\mathbb{R}^{2m} \cong \mathbb{C}^m$. Under this identification, the RoPE block rotation $R_t$ corresponds to multiplication by a diagonal complex matrix $\Lambda(t)$, which both clarifies the structure and motivates the more general complex-diagonal score class considered below:

$$F(x, y, t) = \mathrm{Re}(\langle h(x), \Lambda(t)\, g(y) \rangle_{\mathbb{C}^m}), \qquad \Lambda(t) = \mathrm{diag}(\phi_1(t), \dots, \phi_m(t)), \tag{9}$$

where $h \in C(X, \mathbb{C}^m), g \in C(Y, \mathbb{C}^m)$, and each diagonal entry $\phi_k$ is drawn from a function algebra $\Phi \subset C(T, \mathbb{C})$.

We first show that every real block-rotation score equation 8 (in particular, RoPE) can be written in complex-diagonal form, with $\Lambda(t) = \mathrm{diag}(e^{i\theta_1 t}, \dots, e^{i\theta_m t})$ for suitable fixed frequencies $\theta_k$.

We then prove that, whenever $\Phi$ is a sufficiently rich (self-adjoint) subalgebra of $C(T, \mathbb{C})$, the induced complex-diagonal class is dense in $C(K, \mathbb{R})$ in the uniform norm. This yields the desired universal approximation property; standard RoPE is recovered as a special case by choosing $\Phi$ to contain the relevant exponentials.

**Real-valued block rotations as a complex-diagonal operator.**    Fix $m \in \mathbb{N}$. Define the real-to-complex identification $\rho : \mathbb{R}^{2m} \to \mathbb{C}^m$ by

$$\rho\big((a_1, a_2, \dots, a_{2m-1}, a_{2m})^\top\big) = (a_1 + ia_2,\ a_3 + ia_4,\ \dots,\ a_{2m-1} + ia_{2m})^\top. \tag{10}$$

We use the standard Hermitian inner product on $\mathbb{C}^m$, $\langle u, v \rangle_{\mathbb{C}^m} := u^* v = \sum_{k=1}^m \overline{u_k}\, v_k$. Then for any $a, b \in \mathbb{R}^{2m}$,

$$\langle a, b \rangle_{\mathbb{R}^{2m}} = \mathrm{Re}(\langle \rho(a), \rho(b) \rangle_{\mathbb{C}^m}).$$

Let $\theta_1, \dots, \theta_m \in \mathbb{R}$ be fixed frequencies and let $R_t \in \mathbb{R}^{2m \times 2m}$ be the block-diagonal rotation

$$R_t = \mathrm{diag}\big(R(\theta_1 t), \dots, R(\theta_m t)\big), \qquad R(\alpha) = \begin{pmatrix} \cos \alpha & -\sin \alpha \\ \sin \alpha & \cos \alpha \end{pmatrix}.$$

Define the complex diagonal matrix

$$\Lambda(t) = \mathrm{diag}\big(e^{i\theta_1 t}, \dots, e^{i\theta_m t}\big) \in \mathbb{C}^{m \times m}.$$

A direct check on each $2 \times 2$ block shows that $\rho(R_t z) = \Lambda(t)\rho(z)$ for all $z \in \mathbb{R}^{2m}$. Consequently, for any real-valued encoders $h_{\mathbb{R}} : X \to \mathbb{R}^{2m}$ and $g_{\mathbb{R}} : Y \to \mathbb{R}^{2m}$,

$$\langle h_{\mathbb{R}}(x), R_t g_{\mathbb{R}}(y) \rangle_{\mathbb{R}^{2m}} = \mathrm{Re}(\langle \rho(h_{\mathbb{R}}(x)), \Lambda(t)\, \rho(g_{\mathbb{R}}(y)) \rangle_{\mathbb{C}^m}). \tag{11}$$

---

[2]In Section 3.3 the positional encoding is denoted by $\rho(i)$. Here we avoid this notation to prevent confusion with the map $\rho : \mathbb{R}^{2m} \to \mathbb{C}^m$ used below to identify $\mathbb{R}^{2m} \cong \mathbb{C}^m$.

In particular, whenever an algebra $\Phi \subset C(T, \mathbb{C})$ contains the exponentials $t \mapsto e^{i\theta_k t}$, the real-valued block-rotation score equation 8 (and hence standard RoPE) is a special case of the complex-diagonal score class $\mathcal{A}_\Phi$ defined next.

**Theorem 1** (Complex Inner-product approximation with diagonal positional matrix). *Let $X \subset \mathbb{R}^{d_x}$, $Y \subset \mathbb{R}^{d_y}$ and $T \subset \mathbb{R}$ be compact sets, and $K = X \times Y \times T$. Let $\Phi \subset C(T, \mathbb{C})$ be a* self-adjoint *subalgebra (i.e., $\phi \in \Phi \Rightarrow \overline{\phi} \in \Phi$) which contains constants and separates points of $T$:*

$$1 \in \Phi, \qquad \forall\, t_1 \neq t_2 \,\exists\, \phi \in \Phi: \ \phi(t_1) \neq \phi(t_2).$$

*Define the function class*

$$\mathcal{A}_\Phi = \bigcup_{m \in \mathbb{N}} \left\{ (x, y, t) \mapsto \mathrm{Re}\left( \langle h(x), \Lambda(t) g(y) \rangle_{\mathbb{C}^m} \right) \ \Big|\ h \in C(X, \mathbb{C}^m),\ g \in C(Y, \mathbb{C}^m),\ \Lambda \in \Lambda_{\Phi,m} \right\},$$
(12)

*where*

$$\Lambda_{\Phi,m} := \left\{ \Lambda: T \to \mathbb{C}^{m \times m} \ \Big|\ \Lambda(t) = \mathrm{diag}(\phi_1(t), \ldots, \phi_m(t)),\ \phi_k \in \Phi \right\},$$

*and $\langle u, v \rangle_{\mathbb{C}^m} := u^* v$. Then $\mathcal{A}_\Phi$ is dense in $C(K, \mathbb{R})$ in the uniform norm. Equivalently, for any $f \in C(K, \mathbb{R})$ and any $\varepsilon > 0$ there exist $m$, $h$, $g$ and $\Lambda$ as above such that*

$$\sup_{(x,y,t) \in K} \left| f(x, y, t) - \mathrm{Re}\left( \langle h(x), \Lambda(t) g(y) \rangle_{\mathbb{C}^m} \right) \right| < \varepsilon.$$
(13)

*Proof.* Consider the set

$$\mathcal{B} = \left\{ \sum_{k=1}^J u_k(x)\, v_k(y)\, \phi_k(t) \ \Big|\ J \in \mathbb{N},\ u_k \in C(X, \mathbb{C}),\ v_k \in C(Y, \mathbb{C}),\ \phi_k \in \Phi \right\} \subset C(K, \mathbb{C}).$$

**Density of $\mathcal{B}$ in $C(K, \mathbb{C})$.** First we show that $\mathcal{B}$ is a self-adjoint subalgebra of $C(K, \mathbb{C})$ that contains constants and separates points. Closure under addition and scalar multiplication is immediate. To check closure under products, take

$$b(x, y, t) = \sum_{k=1}^J u_k(x)\, v_k(y)\, \phi_k(t), \qquad b'(x, y, t) = \sum_{\ell=1}^{J'} \tilde{u}_\ell(x)\, \tilde{v}_\ell(y)\, \tilde{\phi}_\ell(t),$$

with $u_k, \tilde{u}_\ell \in C(X, \mathbb{C})$, $v_k, \tilde{v}_\ell \in C(Y, \mathbb{C})$, $\phi_k, \tilde{\phi}_\ell \in \Phi$. Then

$$(b \cdot b')(x, y, t) = \sum_{k=1}^J \sum_{\ell=1}^{J'} \left( u_k \tilde{u}_\ell \right)(x) \left( v_k \tilde{v}_\ell \right)(y) \left( \phi_k \tilde{\phi}_\ell \right)(t).$$

Since $C(X, \mathbb{C})$ and $C(Y, \mathbb{C})$ are algebras under pointwise multiplication, $u_k \tilde{u}_\ell \in C(X, \mathbb{C})$ and $v_k \tilde{v}_\ell \in C(Y, \mathbb{C})$. Because $\Phi$ is a subalgebra, $\phi_k \tilde{\phi}_\ell \in \Phi$. Hence $b \cdot b' \in \mathcal{B}$, so $\mathcal{B}$ is an algebra.

Also $1 \in \mathcal{B}$ by taking $u \equiv 1$, $v \equiv 1$, $\phi \equiv 1$. Self-adjointness follows from pointwise conjugation:

$$\overline{u(x)v(y)\phi(t)} = \overline{u}(x)\, \overline{v}(y)\, \overline{\phi}(t),$$

and the assumptions $\overline{\phi} \in \Phi$ and $\overline{u} \in C(X, \mathbb{C})$, $\overline{v} \in C(Y, \mathbb{C})$.

Additionally $\mathcal{B}$ separates points of $K$. Let $(x_1, y_1, t_1) \neq (x_2, y_2, t_2)$. If $x_1 \neq x_2$, choose a coordinate projection $u(x) = x_j$ with $x_{1,j} \neq x_{2,j}$, and set $v \equiv 1$, $\phi \equiv 1$; then $u(x_1) \neq u(x_2)$. Similarly if $y_1 \neq y_2$. If $t_1 \neq t_2$, use the assumption that $\Phi$ separates points of $T$: pick $\phi \in \Phi$ with $\phi(t_1) \neq \phi(t_2)$ and set $u \equiv 1$, $v \equiv 1$.

By the complex Stone–Weierstrass theorem (self-adjoint subalgebra, contains constants, separates points), $\overline{\mathcal{B}} = C(K, \mathbb{C})$.

**Real parts of $\mathcal{B}$ lie in $\mathcal{A}_\Phi$.** Take any $b \in \mathcal{B}$:

$$b(x,y,t) = \sum_{k=1}^{J} u_k(x)\, v_k(y)\, \phi_k(t).$$

Let $m = J$,

$$h(x) = (\overline{u_1(x)}, \dots, \overline{u_J(x)})^\top, \quad g(y) = (v_1(y), \dots, v_J(y))^\top, \quad \Lambda(t) = \mathrm{diag}(\phi_1(t), \dots, \phi_J(t)).$$

Using $\langle a, b \rangle_{\mathbb{C}^m} = a^* b$,

$$\langle h(x), \Lambda(t) g(y) \rangle_{\mathbb{C}^m} = \sum_{k=1}^{J} \overline{h_k(x)}\, \phi_k(t)\, g_k(y) = \sum_{k=1}^{J} u_k(x)\, v_k(y)\, \phi_k(t) = b(x,y,t), \qquad (14)$$

hence $\mathrm{Re}(b) \in \mathcal{A}_\Phi$.

**Approximation of real-valued functions.** Let $f \in C(K, \mathbb{R})$ and $\varepsilon > 0$. View $f$ as an element of $C(K, \mathbb{C})$. Pick $b \in \mathcal{B}$ such that $\|f - b\|_\infty < \varepsilon$. Define $F = \mathrm{Re}(b)$. Then, using $|\mathrm{Re}\, z| \le |z|$,

$$\|f - F\|_\infty = \|f - \mathrm{Re}(b)\|_\infty = \|\mathrm{Re}(f - b)\|_\infty \le \|f - b\|_\infty < \varepsilon. \qquad (15)$$

As shown above $F \in \mathcal{A}_\Phi$, which proves density in $C(K, \mathbb{R})$. $\qquad \square$

The qualitative approximation property established in Theorem 1 does not reveal how the required feature dimension scales with the desired accuracy. A first step toward a quantitative understanding is Lemma 1, which gives a rate $d^{-s/(d_x+d_y)}$ for approximating stationary kernels in $H^s(X \times Y)$. By incorporating this lemma into a Fourier analysis of the temporal variable, we obtain the following theorem that handles the full $(x,y,t)$-dependent case and achieves the rate $d^{-s/(d_x+d_y+1)}$.

**Lemma 1** ($L^2$ low-rank approximation of Sobolev kernels). *Let $X \subset \mathbb{R}^{d_x}$ and $Y \subset \mathbb{R}^{d_y}$ be bounded Lipschitz domains, and set $D = d_x + d_y$. Let $s > 0$ and let $a \in H^s(X \times Y)$ be real(or complex)-valued. Then for every integer $d \ge 1$ there exist measurable feature maps*

$$h \in L^2(X; \mathbb{C}^d), \qquad g \in L^2(Y; \mathbb{C}^d), \qquad (16)$$

*such that*

$$\left\| a(\cdot, \cdot) - \langle h(\cdot), g(\cdot) \rangle_{\mathbb{C}^d} \right\|_{L^2(X \times Y)} \le C\, d^{-s/D} \|a\|_{H^s(X \times Y)}. \qquad (17)$$

*Here $C > 0$ depends only on $s, d_x, d_y$ and the geometry of $X, Y$.*

*Proof.* Consider the bounded Lipschitz domain $\Omega := X \times Y \subset \mathbb{R}^D$. By a Sobolev extension theorem for bounded Lipschitz domains, there exists a bounded linear operator

$$E : H^s(\Omega) \to H^s(\mathbb{R}^D) \qquad (18)$$

such that $Eu|_\Omega = u$ for all $u \in H^s(\Omega)$. Apply it to $a$: let $\widetilde{a} := Ea \in H^s(\mathbb{R}^D)$, then

$$\|\widetilde{a}\|_{H^s(\mathbb{R}^D)} \le C_{\mathrm{ext}} \|a\|_{H^s(\Omega)}. \qquad (19)$$

Choose a cube $Q \subset \mathbb{R}^D$ with side length $L$ such that $\overline{\Omega} \subset Q$. Take a cut-off function $\chi \in C_c^\infty(Q)$ satisfying $\chi \equiv 1$ on $\Omega$ and $0 \le \chi \le 1$, with $\mathrm{supp}\,\chi \Subset Q$ (in particular, $\mathrm{dist}(\mathrm{supp}\,\chi, \partial Q) > 0$). Define $u := \chi \cdot \widetilde{a}$. Then $u \in H^s(\mathbb{R}^D)$, $\mathrm{supp}\, u \subset Q$, and $u|_\Omega = a$. By boundedness of multiplication by a smooth compactly supported function,

$$\|u\|_{H^s(\mathbb{R}^D)} \le C_\chi \|\widetilde{a}\|_{H^s(\mathbb{R}^D)} \le C_\chi C_{\mathrm{ext}} \|a\|_{H^s(\Omega)}. \qquad (20)$$

Now periodise $u$. Since $\mathrm{supp}\, u$ is compactly contained in $Q$, the function $u$ vanishes in a neighbourhood of $\partial Q$. Identify $Q$ with a fundamental domain of the $D$-dimensional torus $\mathbb{T}_L^D := \mathbb{R}^D / (L\mathbb{Z})^D$. Define the $L$-periodic extension

$$u_{\mathrm{per}}(z) := \sum_{k \in \mathbb{Z}^D} u(z + Lk), \quad z \in \mathbb{R}^D. \qquad (21)$$

The sum is locally finite and defines an $L$-periodic function on $\mathbb{R}^D$. Because $u$ vanishes near $\partial Q$, there is no jump across the faces of $Q$; hence $u_{\mathrm{per}}$ defines a function on $\mathbb{T}_L^D$. Moreover, the standard estimate for periodisation yields

$$\|u_{\mathrm{per}}\|_{H^s(\mathbb{T}_L^D)} \leq C_{\mathrm{per}}\|u\|_{H^s(\mathbb{R}^D)}, \tag{22}$$

where $C_{\mathrm{per}}$ depends only on $L$ (equivalently, on the choice of $Q$).

On $\mathbb{T}_L^D$ the $H^s$-norm can be expressed via Fourier coefficients:

$$\|v\|_{H^s(\mathbb{T}_L^D)}^2 \asymp_L \sum_{k\in\mathbb{Z}^D} (1+|k|^2)^s |\widehat{v}_k|^2, \tag{23}$$

where $\widehat{v}_k$ are the Fourier coefficients in the expansion

$$u_{\mathrm{per}}(z) = \sum_{k\in\mathbb{Z}^D} \widehat{u}_k\, e^{2\pi i k\cdot z/L}, \qquad z = (x,y) \in \mathbb{T}_L^D. \tag{24}$$

For a given integer $N \geq 1$ consider the truncated sum

$$P_N(z) := \sum_{\|k\|_\infty \leq N} \widehat{u}_k\, e^{2\pi i k\cdot z/L}, \tag{25}$$

a trigonometric polynomial of degree $N$. By Parseval's identity,

$$\|u_{\mathrm{per}} - P_N\|_{L^2(\mathbb{T}_L^D)}^2 = \sum_{\|k\|_\infty > N} |\widehat{u}_k|^2. \tag{26}$$

For $\|k\|_\infty > N$ we have $|k| \geq N$, hence $(1+|k|^2)^{-s} \leq (1+N^2)^{-s}$, so
$$|\widehat{u}_k|^2 \leq (1+N^2)^{-s}(1+|k|^2)^s |\widehat{u}_k|^2.$$
Inserting this into equation 26 and using equation 23 gives

$$\|u_{\mathrm{per}} - P_N\|_{L^2(\mathbb{T}_L^D)}^2 \leq C(1+N^2)^{-s}\|u_{\mathrm{per}}\|_{H^s(\mathbb{T}_L^D)}^2. \tag{27}$$

Taking square roots and using $(1+N^2)^{-s/2} \leq C'N^{-s}$ yields

$$\|u_{\mathrm{per}} - P_N\|_{L^2(\mathbb{T}_L^D)} \leq C_1 N^{-s}\|u_{\mathrm{per}}\|_{H^s(\mathbb{T}_L^D)}. \tag{28}$$

Combining equation 19, equation 20, equation 22, and equation 28 we obtain

$$\|u_{\mathrm{per}} - P_N\|_{L^2(\mathbb{T}_L^D)} \leq C_2 N^{-s}\|a\|_{H^s(\Omega)}. \tag{29}$$

Since $u_{\mathrm{per}} = u = a$ on $\Omega$, restriction does not increase the $L^2$-error:

$$\|a - P_N\|_{L^2(\Omega)} \leq \|u_{\mathrm{per}} - P_N\|_{L^2(\mathbb{T}_L^D)} \leq C_2 N^{-s}\|a\|_{H^s(\Omega)}. \tag{30}$$

Now count the number of retained modes:

$$d := \#\{k \in \mathbb{Z}^D : \|k\|_\infty \leq N\} = (2N+1)^D \asymp N^D. \tag{31}$$

Hence $N^{-s} \asymp d^{-s/D}$, and equation 30 implies

$$\|a - P_N\|_{L^2(\Omega)} \leq C_3 d^{-s/D}\|a\|_{H^s(\Omega)}. \tag{32}$$

It remains to write $P_N$ as an inner product of feature maps. Write $k = (k_x, k_y)$ with $k_x \in \mathbb{Z}^{d_x}$, $k_y \in \mathbb{Z}^{d_y}$, so that $e^{2\pi i k\cdot(x,y)/L} = e^{2\pi i k_x\cdot x/L}\, e^{2\pi i k_y\cdot y/L}$. Enumerate all admissible indices $k$ with $\|k\|_\infty \leq N$ as $k^{(1)},\ldots,k^{(d)}$ and set

$$h_j(x) := \overline{\widehat{u}_{k^{(j)}}}\, e^{-2\pi i k_x^{(j)}\cdot x/L}, \qquad g_j(y) := e^{2\pi i k_y^{(j)}\cdot y/L}. \tag{33}$$

Define the vector-valued maps $h(x) = (h_1(x),\ldots,h_d(x))$ and $g(y) = (g_1(y),\ldots,g_d(y))$. Then $h \in L^2(X;\mathbb{C}^d)$ and $g \in L^2(Y;\mathbb{C}^d)$, and using the standard Hermitian inner product $\langle \alpha, \beta\rangle_{\mathbb{C}^d} := \sum_{j=1}^d \overline{\alpha_j}\beta_j$ we obtain

$$\langle h(x), g(y)\rangle_{\mathbb{C}^d} = \sum_{j=1}^d \widehat{u}_{k^{(j)}} e^{2\pi i(k_x^{(j)}\cdot x + k_y^{(j)}\cdot y)/L} = P_N(x,y). \tag{34}$$

Together with equation 32 this yields the claim. The constant $C_3$ depends only on $s$, $d_x$, $d_y$, and the geometry of $X, Y$ (through the norms of the extension, cut-off, and periodisation operators and the choice of $Q$). $\qquad\square$

**Theorem 2** (Approximation by RoPE-type feature maps in $L^2$). *Let $X \subset \mathbb{R}^{d_x}$ and $Y \subset \mathbb{R}^{d_y}$ be bounded Lipschitz domains, set $D = d_x + d_y$, and let $T = \mathbb{R}/\{2\pi\mathbb{Z}\}$ with endpoints identified.*

*Let $s > 0$ be an integer, consider a real-valued function $f$ and assume*

$$f \in L^2\big(T; H^s(X \times Y)\big), \quad \partial_t^s f \in L^2\big(T; L^2(X \times Y)\big). \tag{35}$$

*Define*

$$M := \|f\|_{L^2(T; H^s(X \times Y))} + \|\partial_t^s f\|_{L^2(T; L^2(X \times Y))}. \tag{36}$$

*Then there exists a constant $C > 0$, depending only on $s, d_x, d_y$ and the geometry of $X, Y$, such that for every integer $d \geq 1$ one can find feature maps $h \in L^2(X; \mathbb{C}^{d'})$ and $g \in L^2(Y; \mathbb{C}^{d'})$ with $d' \leq d$, and a family of unitary matrices $\{\Lambda(t)\}_{t \in T} \subset \mathbb{C}^{d' \times d'}$ of the form*

$$\Lambda(t) = \mathrm{diag}\big(e^{i\omega_1 t}, \ldots, e^{i\omega_{d'} t}\big), \qquad \omega_j \in \mathbb{Z}, \tag{37}$$

*such that*

$$\big\| f(\cdot, \cdot, \cdot) - \mathrm{Re}\left(\langle h(\cdot), \Lambda(\cdot)g(\cdot)\rangle_{\mathbb{C}^{d'}}\right) \big\|_{L^2(X \times Y \times T)} \leq C \, M \, d^{-\beta}, \qquad \beta = \frac{s}{D+1}. \tag{38}$$

*Proof.* We begin by expanding $f$ in a Fourier series with respect to the periodic variable $t \in T$. Define the Fourier coefficients

$$a_k(x, y) := \frac{1}{2\pi} \int_0^{2\pi} f(x, y, t) \, e^{-ikt} \, dt, \qquad k \in \mathbb{Z}, \tag{39}$$

which belong to $L^2(X \times Y)$ because $f \in L^2(T; L^2(X \times Y))$. Then $f$ can be written as

$$f(x, y, t) = \sum_{k \in \mathbb{Z}} a_k(x, y)e^{ikt}, \tag{40}$$

and Parseval's identity gives the equalities

$$\|f\|_{L^2(X \times Y \times T)}^2 = 2\pi \sum_{k \in \mathbb{Z}} \|a_k\|_{L^2(X \times Y)}^2, \tag{41}$$

$$\|\partial_t^s f\|_{L^2(X \times Y \times T)}^2 = 2\pi \sum_{k \in \mathbb{Z}} |k|^{2s} \|a_k\|_{L^2(X \times Y)}^2. \tag{42}$$

These relations are the starting point for both the temporal truncation and the low-rank spatial approximation.

**Temporal truncation.** For a cut-off frequency $N \in \mathbb{N}$ we consider the truncated Fourier sum

$$f^{(N)}(x, y, t) := \sum_{|k| \leq N} a_k(x, y)e^{ikt}. \tag{43}$$

Using equation 41 and equation 42 we estimate the error made by this truncation:

$$\begin{aligned}
\|f - f^{(N)}\|_{L^2(X \times Y \times T)}^2 &= 2\pi \sum_{|k| > N} \|a_k\|_{L^2}^2 \\
&\leq 2\pi N^{-2s} \sum_{|k| > N} |k|^{2s} \|a_k\|_{L^2}^2 \\
&\leq N^{-2s} \|\partial_t^s f\|_{L^2(X \times Y \times T)}^2.
\end{aligned} \tag{44}$$

Taking square roots we obtain

$$\|f - f^{(N)}\|_{L^2(X \times Y \times T)} \leq N^{-s} \|\partial_t^s f\|_{L^2(X \times Y \times T)}. \tag{45}$$

**Low-rank approximation of the spatial coefficients.** Because $f \in L^2(T; H^s(X \times Y))$, each coefficient $a_k$ belongs to $H^s(X \times Y)$ and the $H^s$ norms satisfy the Parseval-type relation

$$2\pi \sum_{k \in \mathbb{Z}} \|a_k\|_{H^s(X \times Y)}^2 = \|f\|_{L^2(T; H^s(X \times Y))}^2. \tag{46}$$

We now fix an integer $r \geq 1$ (the same for all frequencies $|k| \leq N$) and apply Lemma 1 to every $a_k$ with rank parameter $d = r$. The lemma supplies functions $h_k : X \to \mathbb{C}^r$ and $g_k : Y \to \mathbb{C}^r$ such that

$$\|a_k - \langle h_k, g_k \rangle\|_{L^2(X \times Y)} \leq C_0 \, r^{-s/D} \|a_k\|_{H^s(X \times Y)}. \tag{47}$$

Using these approximations we define a function that mimics $f^{(N)}$:

$$\widetilde{f}^{(N)}(x, y, t) := \sum_{|k| \leq N} e^{ikt} \langle h_k(x), g_k(y) \rangle_{\mathbb{C}^r}. \tag{48}$$

Because the Fourier modes $e^{ikt}$ are orthogonal in $L^2(T)$, the error between $f^{(N)}$ and $\widetilde{f}^{(N)}$ can be expressed without cross-terms:

$$\|f^{(N)} - \widetilde{f}^{(N)}\|_{L^2(X \times Y \times T)}^2 = 2\pi \sum_{|k| \leq N} \|a_k - \langle h_k, g_k \rangle\|_{L^2(X \times Y)}^2. \tag{49}$$

Inserting the estimate equation 47 and using equation 46 we obtain

$$\|f^{(N)} - \widetilde{f}^{(N)}\|_{L^2(X \times Y \times T)} \leq C_1 \, r^{-s/D} \|f\|_{L^2(T; H^s(X \times Y))}, \tag{50}$$

where $C_1$ is a constant that absorbs $C_0$ and the factor coming from the sum of the squared $H^s$ norms.

**Assembling a global RoPE-type representation.** Let us now collect all the component maps into single vectors. Set

$$d' := (2N + 1)r, \tag{51}$$

and note that $d' \leq d$ by the choice $r = \lfloor d/(2N + 1) \rfloor$. Define

$$h(x) := \big(h_{-N}(x), \ldots, h_N(x)\big) \in \mathbb{C}^{d'}, \quad g(y) := \big(g_{-N}(y), \ldots, g_N(y)\big) \in \mathbb{C}^{d'}. \tag{52}$$

For the rotation matrices we take the block-diagonal operator

$$\Lambda(t) := \mathrm{diag}\big(e^{-iNt} I_r, \ e^{-i(N-1)t} I_r, \ \ldots, \ e^{iNt} I_r\big), \tag{53}$$

which is clearly of the form $\mathrm{diag}(e^{i\omega_1 t}, \ldots, e^{i\omega_{d'} t})$ with integer frequencies $\omega_j$ (each frequency $k$ is repeated $r$ times). A direct computation shows that

$$\langle h(x), \Lambda(t) g(y) \rangle_{\mathbb{C}^{d'}} = \sum_{|k| \leq N} e^{ikt} \langle h_k(x), g_k(y) \rangle_{\mathbb{C}^r} = \widetilde{f}^{(N)}(x, y, t). \tag{54}$$

**Choice of the cut-off $N$ and final error estimate.** Combining the estimates equation 45, equation 50 and the identity equation 54 we obtain from the triangle inequality

$$\|f - \langle h, \Lambda(t) g \rangle\|_{L^2} \leq \|f - f^{(N)}\|_{L^2} + \|f^{(N)} - \widetilde{f}^{(N)}\|_{L^2} \tag{55}$$

$$\leq N^{-s} \|\partial_t^s f\|_{L^2} + C_1 r^{-s/D} \|f\|_{L^2(T; H^s)}. \tag{56}$$

Recall that $r$ is related to the total dimension $d$ by $r = \lfloor d/(2N + 1) \rfloor$; for large $d$ we have $r \asymp d/N$. Substituting this asymptotic relation into equation 55 yields

$$\|f - \langle h, \Lambda(t) g \rangle\|_{L^2} \leq C_2 \Big(N^{-s} \|\partial_t^s f\|_{L^2} + (d/N)^{-s/D} \|f\|_{L^2(T; H^s)}\Big).$$

We now choose $N$ so that the two terms balance. Setting $N = \lfloor d^{1/(D+1)} \rfloor$ gives

$$N^{-s} \asymp d^{-s/(D+1)}, \qquad (d/N)^{-s/D} \asymp d^{-s/(D+1)}.$$

Consequently,

$$\|f - \langle h, \Lambda(t) g \rangle\|_{L^2(X \times Y \times T)} \leq C \Big(\|\partial_t^s f\|_{L^2} + \|f\|_{L^2(T; H^s)}\Big) d^{-s/(D+1)}. \tag{57}$$

Finally, note that the quantity $\|\partial_t^s f\|_{L^2} + \|f\|_{L^2(T;H^s)}$ is bounded by a constant multiple of the norm $M$ appearing in the statement of the theorem (because the $L^2$ norms are controlled by the corresponding $C(T;H^s)$ norms). Thus we arrive at the desired estimate

$$\|f - \mathrm{Re}(\langle h, \Lambda(t)g\rangle)\|_{L^2} \le \|f - \langle h, \Lambda(t)g\rangle\|_{L^2} \le C\, M\, d^{-s/(D+1)}. \tag{58}$$

$\square$

**Remark.** Theorem 2 provides an $L^2(X \times Y \times T)$ error bound, which is natural for mean-square losses and leads to a clean rate because orthogonality in $t$ (Parseval) can be fully exploited. A uniform-in-$(x, y, t)$ bound,

$$\|f - \langle h, \Lambda(\cdot)g\rangle\|_{L^\infty(X \times Y \times T)}, \tag{59}$$

is strictly stronger and requires additional ingredients beyond the $L^2$ proof. There are two standard routes.

**1. Uniform bounds from Sobolev embedding (typically slower rates under $H^s$ only).** Assume only the spatial Sobolev regularity used in Theorem 2, namely $f(\cdot, \cdot, t) \in H^s(X \times Y)$ with $s > D/2$. To pass from an $H^s$-estimate to $L^\infty$ one uses Sobolev embedding $H^{s_0}(X \times Y) \hookrightarrow L^\infty(X \times Y)$ with any $s_0 > D/2$. A typical spectral-truncation argument then gives, for any fixed $s_0 \in (D/2, s)$, a low-rank bound of the form

$$\|a - \mathrm{rank}\text{-}d\|_{L^\infty(X \times Y)} \le C\, d^{-(s-s_0)/D}\, \|a\|_{H^s(X \times Y)}. \tag{60}$$

Choosing $s_0 = D/2 + \varepsilon$ yields the more explicit (but slightly weaker) rate

$$\|a - \mathrm{rank}\text{-}d\|_{L^\infty(X \times Y)} \le C_\varepsilon\, d^{-(s-D/2-\varepsilon)/D}\, \|a\|_{H^s(X \times Y)}. \tag{61}$$

Plugging this $L^\infty$ low-rank estimate into the RoPE construction (and using a vector-valued Jackson bound in $t$ followed by Sobolev embedding in $(x, y)$) leads to a uniform approximation rate

$$\|f - \langle h, \Lambda(\cdot)g\rangle\|_{L^\infty(X \times Y \times T)} \le C_\varepsilon\, M\, d^{-\beta_\varepsilon}, \qquad \beta_\varepsilon = \frac{s(s - D/2 - \varepsilon)}{Ds + s - D/2 - \varepsilon}. \tag{62}$$

In particular, one generally pays an $\varepsilon$-loss and a smaller exponent than in the $L^2$ theorem when only $H^s$-regularity is assumed in space.

**2. Recovering clean $L^\infty$ rates by strengthening spatial smoothness.** If one strengthens the spatial regularity of $f$ (uniformly in $t$) so that an $L^\infty$-stable approximation theory applies, then the uniform RoPE approximation can match the clean algebraic rates. One convenient sufficient condition is additional Sobolev smoothness: assume that for some $\sigma > 0$,

$$\partial_t^\ell f \in C\big(T; H^{s+\sigma}(X \times Y)\big), \qquad 0 \le \ell \le s, \tag{63}$$

and retain $s > D/2$ to guarantee $H^s(X \times Y) \hookrightarrow L^\infty(X \times Y)$. Then one can approximate each Fourier coefficient $a_k$ in $H^s$ at rate $r^{-\sigma/D}\|a_k\|_{H^{s+\sigma}}$, and after embedding obtain

$$\|f - \langle h, \Lambda(\cdot)g\rangle\|_{L^\infty(X \times Y \times T)} \le C\, M_\sigma\Big(N^{-s} + r^{-\sigma/D}\Big), \tag{64}$$

with $M_\sigma := \max_{0 \le \ell \le s} \|\partial_t^\ell f\|_{C(T;H^{s+\sigma})}$ and $r \approx d/(2N+1)$ as in the proof. Optimizing $N$ then yields an exponent

$$\beta = \frac{s\sigma}{Ds + \sigma}, \tag{65}$$

and in the balanced case $\sigma = s$ this recovers $\beta = s/(D+1)$ in $L^\infty$ (at the cost of requiring $H^{2s}$-type smoothness in $(x, y)$).

Theorem 2 only requires $h \in L^2(X; \mathbb{C}^d)$ and $g \in L^2(Y; \mathbb{C}^d)$. Uniform-in-$(x, y)$ bounds typically require additional regularity (and sometimes stronger boundary regularity of $X, Y$) if one wants $h, g$ to be continuous on $\overline{X}, \overline{Y}$.

The $L^2$ theorem is sharp and technically robust under minimal assumptions. Uniform $L^\infty$ control is stronger, but it either yields a slower rate under pure $H^s$ assumptions or requires stronger spatial smoothness assumptions to retain the clean exponent.

## B  EARLY STOPPING EXPERIMENTS

In this section, we study a simple early stopping rule for the retrieval agent. Let

$$\mathbf{a} = (a_1, a_2, \ldots, a_T)$$

be the full sequence of chunks the agent would select if no stopping threshold were applied, and let $G$ be a set of ground-truth chunks for the current question.

For each step $t$, the agent outputs a Q-value $Q_t$ for taking the next retrieval action. Given a fixed Q-value threshold $Q_{\text{threshold}}$, we simulate an early-stopping policy that keeps taking actions while $Q_t \geq Q_{\text{threshold}}$ and terminates as soon as $Q_t < Q_{\text{threshold}}$. We denote by $t_{\text{stop}}$ the number of actions actually taken under this policy, i.e. the number of selected chunks:

$$t_{\text{stop}} = \text{number of steps until the first } t \text{ with } Q_t < Q_{\text{threshold}}.$$

Independently of the stopping rule, we define $t_{\text{earliest}}$ as the earliest step at which all ground-truth chunks have already been collected:

$$t_{\text{earliest}} = \min\{t \; : \; \{a_1, \ldots, a_t\} \supseteq G\}.$$

If the agent never collects all ground-truth chunks, i.e. such a $t$ does not exist, we discard this episode from the analysis below.

For comparison, we also consider an oracle stopping policy that is allowed to look at the ground truth: it knows $t_{\text{earliest}}$ for each episode and simply stops at this step. By construction, this oracle policy never stops too early or too late.

Depending on the relation between $t_{\text{stop}}$ and $t_{\text{earliest}}$ we distinguish three outcomes.

**Early stop ("early").**  If $t_{\text{stop}} < t_{\text{earliest}}$, the stopping rule terminates before all ground-truth chunks have been selected. In this case the error is due to stopping too early and missing potentially useful chunks.

**Perfect stop ("perfect").**  If $t_{\text{stop}} = t_{\text{earliest}}$, the stopping rule terminates exactly at the first step when the set of selected chunks already contains all ground-truth chunks. In this case, the stopping behavior is optimal with respect to our definition.

**Late stop ("late").**  If $t_{\text{stop}} > t_{\text{earliest}}$, then at some earlier step the agent had already collected all ground-truth chunks but continued to retrieve additional chunks. This corresponds to stopping too late and taking unnecessary steps.

Figure 4 (top row, panel (a)) shows how the proportions of early and late errors change as a function of the Q-value threshold $Q_{\text{threshold}}$ on HotPotQA. For small thresholds, the agent almost never stops too early but may continue to retrieve redundant chunks, which leads to late errors. As the threshold increases, late errors decrease, but the probability of stopping too early grows.

Panel (b) of Figure 4 reports the proportion of "perfect" stopping events, peaking around thresholds $Q_{\text{threshold}} \approx 0.1$–$0.3$. Panel (c) shows the average number of selected chunks (episode length) under the same policy. Larger thresholds lead to shorter episodes, but once the threshold becomes too high, the early-stop error rate rapidly increases and performance degrades.

 Table5 summarises these trade-offs quantitatively on HotPotQA for the GTE embedder with `penalize_extra_steps=True` and `never_terminate=True`. We report the fraction of early, late and perfect stops, the average episode length, and the final Fact EM and Fact F1 scores, as well as the corresponding true positive rate (TPR) and false positive rate (FPR) for the stopping rule viewed as a binary classifier. The best Fact F1 is achieved at $Q_{\text{threshold}} = 0.2$, confirming that moderate thresholds provide a good balance between taking enough retrieval steps and avoiding unnecessary ones.

Using the TPR and FPR columns of Tables 5 and 6, we can plot the receiver operating characteristic (ROC) curves of the early-stopping rule, shown in Figure 5. Panel (a) corresponds to HotPotQA and panel (b) to BabiLong QA2. Each point on the curves corresponds to a particular $Q$-value threshold

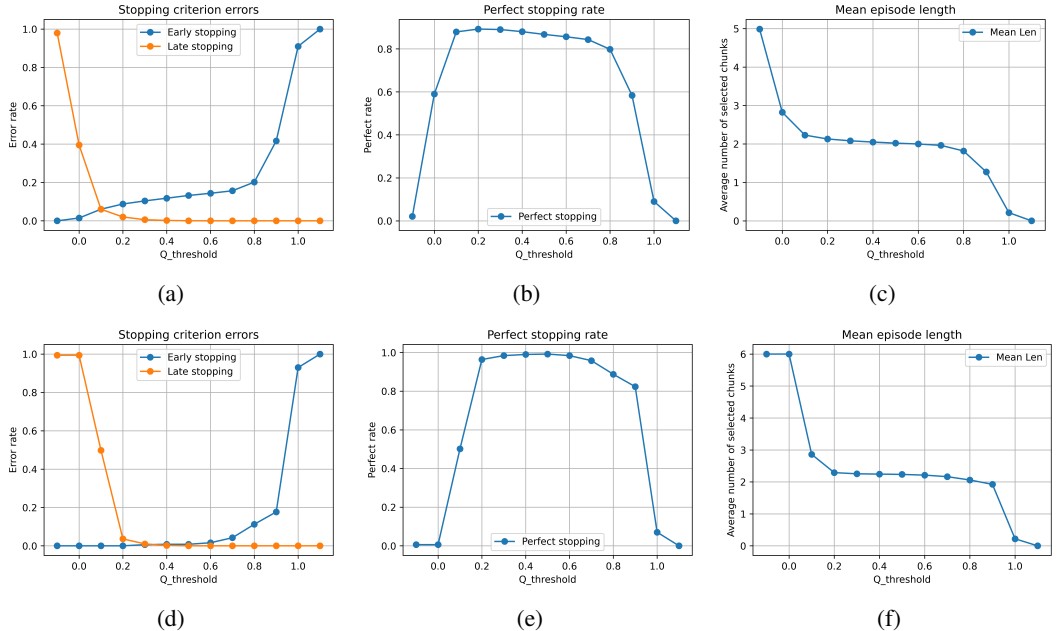

Figure 4: Early stopping analysis on HotPotQA (top row) and BabiLong QA2 (bottom row). Panels (a,d) show the proportions of early and late errors as a function of the Q-value threshold $Q_{\text{threshold}}$. Panels (b,e) show the proportion of perfect stops. Panels (c,f) show the average number of selected chunks (episode length).

Table 5: HotPotQA early stopping experiments

| $Q$-value threshold | stopped early | stopped later | perfect stop | TPR | FPR | Episode len | Fact EM | Fact F1 | Ans EM | Ans F1 |
|---|---|---|---|---|---|---|---|---|---|---|
| -0.1 | 0 | 0.979 | 0.021 | 0.983 | 0.380 | 4.99 | 0.968 | 0.563 | 0.588 | 0.759 |
| 0.0 | 0.015 | 0.395 | 0.590 | 0.976 | 0.110 | 2.82 | 0.954 | 0.843 | 0.592 | 0.761 |
| 0.1 | 0.061 | 0.060 | 0.879 | 0.952 | 0.041 | 2.23 | 0.910 | 0.915 | 0.593 | 0.756 |
| 0.2 | 0.088 | 0.020 | 0.892 | 0.937 | 0.032 | 2.13 | 0.883 | 0.917 | 0.587 | 0.752 |
| 0.3 | 0.104 | 0.006 | 0.890 | 0.927 | 0.029 | 2.08 | 0.868 | 0.915 | 0.585 | 0.747 |
| 0.4 | 0.118 | 0.002 | 0.880 | 0.919 | 0.027 | 2.05 | 0.854 | 0.911 | 0.575 | 0.737 |
| 0.5 | 0.132 | 0 | 0.867 | 0.910 | 0.025 | 2.02 | 0.840 | 0.907 | 0.571 | 0.734 |
| 0.6 | 0.144 | 0 | 0.856 | 0.903 | 0.024 | 2.00 | 0.829 | 0.902 | 0.570 | 0.730 |
| 0.7 | 0.157 | 0 | 0.843 | 0.891 | 0.023 | 1.96 | 0.817 | 0.895 | 0.564 | 0.724 |
| 0.8 | 0.202 | 0 | 0.798 | 0.840 | 0.017 | 1.82 | 0.773 | 0.847 | 0.546 | 0.702 |
| 0.9 | 0.417 | 0 | 0.583 | 0.611 | 0.006 | 1.27 | 0.565 | 0.620 | 0.444 | 0.588 |
| 1.0 | 0.910 | 0 | 0.090 | 0.105 | 0.000 | 0.21 | 0.088 | 0.111 | 0.266 | 0.385 |
| 1.1 | 1.000 | 0 | 0 | 0 | 0 | 0 | 0 | 0 | – | – |

$Q_{\text{threshold}}$. The red star in each panel marks the oracle stopping policy introduced above, which knows $t_{\text{earliest}}$ and stops exactly at that step; this point serves as an upper bound on the achievable trade-off between TPR and FPR. On HotPotQA the area under the curve (AUC) is $0.96$, and for BabiLong QA2 it is $0.97$.

Figure 4 (bottom row) and Table 6 report the same analysis on BabiLong QA2. Qualitatively, the behaviour of the stopping rule is similar to HotPotQA: higher thresholds lead to shorter episodes and more early stops, while lower thresholds reduce early-stop errors at the cost of more late stops and longer episodes.

However, the transition between these regimes is much sharper on BabiLong QA2. For thresholds in the range $Q_{\text{threshold}} \in [0.2, 0.6]$ the fraction of perfect stops remains very high ($\approx 0.95$–$0.99$), while the average episode length is reduced from about $6$ to roughly $2.2$ retrieval steps. In this region Fact EM and Fact F1 stay close to their maximum values (Fact F1 around $0.95$), and answer accuracy (Ans EM/F1) is also near-optimal. Only when the threshold approaches $1.0$, performance collapses, as the agent stops almost immediately and misses relevant chunks.

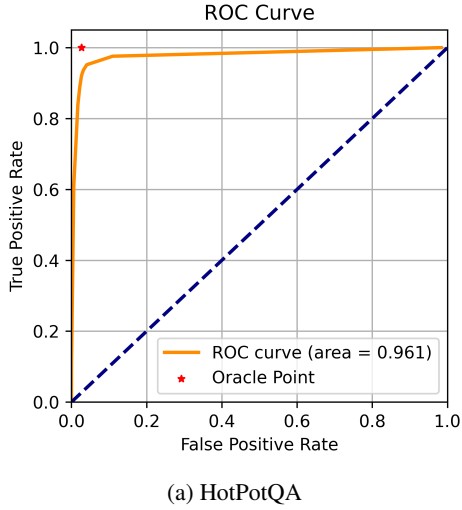
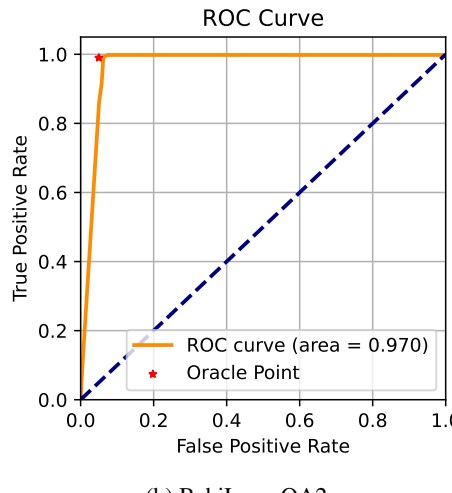

(a) HotPotQA

(b) BabiLong QA2

Figure 5: ROC curves for the early-stopping rule. Panel (a) shows HotPotQA; panel (b) shows Ba-biLong QA2. The dashed line indicates random performance. Each point corresponds to a different $Q$-value threshold $Q_{\text{threshold}}$. The red star denotes the oracle stopping policy that always stops at $t_{\text{earliest}}$, i.e. exactly when the last ground-truth chunk has been retrieved.

Table 6: BabiLong QA2 early stopping experiments.

| $Q$-value threshold | stopped early | stopped later | perfect stop | Episode len | Fact EM | Fact F1 | Ans EM | Ans F1 |
|---|---|---|---|---|---|---|---|---|
| -0.10 | 0.000 | 0.994 | 0.006 | 6.00 | 0.996 | 0.499 | 0.884 | 0.884 |
| 0.00 | 0.000 | 0.994 | 0.006 | 6.00 | 0.996 | 0.499 | 0.884 | 0.884 |
| 0.10 | 0.000 | 0.498 | 0.502 | 2.86 | 0.996 | 0.845 | 0.944 | 0.944 |
| 0.20 | 0.000 | 0.036 | 0.964 | 2.29 | 0.996 | 0.949 | 0.976 | 0.976 |
| 0.30 | 0.006 | 0.010 | 0.984 | 2.25 | 0.990 | 0.952 | 0.970 | 0.970 |
| 0.40 | 0.008 | 0.002 | 0.990 | 2.24 | 0.988 | 0.953 | 0.970 | 0.970 |
| 0.50 | 0.008 | 0.000 | 0.992 | 2.23 | 0.988 | 0.954 | 0.972 | 0.972 |
| 0.60 | 0.016 | 0.000 | 0.984 | 2.21 | 0.980 | 0.948 | 0.968 | 0.968 |
| 0.70 | 0.042 | 0.000 | 0.958 | 2.16 | 0.954 | 0.934 | 0.948 | 0.948 |
| 0.80 | 0.112 | 0.000 | 0.888 | 2.06 | 0.884 | 0.905 | 0.884 | 0.884 |
| 0.90 | 0.177 | 0.000 | 0.823 | 1.92 | 0.820 | 0.861 | 0.830 | 0.830 |
| 1.00 | 0.930 | 0.000 | 0.070 | 0.22 | 0.070 | 0.107 | 0.230 | 0.230 |
| 1.10 | 1.000 | 0.000 | 0.000 | 0.00 | 0.000 | 0.000 | 0.000 | 0.000 |

## C  PLANNING FOR MULTI-STEP RETRIEVAL

We *can* apply **planning** at the multi-step retrieval stage, formulating source selection as a search over the space of action trajectories; see § 4.4 for an application. In the spirit of *Beam Retriever*, we can run beam search where candidates are ranked by the learned action-value function $Q_\theta(s, a)$. However, our planning is computationally cheaper because $Q_\theta$ is computed as a *dot product* of state and action embeddings, $Q_\theta(s, a) = \langle E_s(s), E_a(a) \rangle$, so no new transformer forward passes are required for each candidate chunk, whereas *Beam Retriever* relies on a transformer reranker over trajectories, incurring fresh forward passes at every expansion. Details of the embedding-based scoring are provided in § 3.2. At inference, we perform *beam search over Q* and *deterministically* expand the top-$k$ actions by $Q_\theta$.

## D  METHOD COMPLEXITY AND EFFICIENCY

Q-RAG produces a final answer using two main components. The first is a *multi-step retrieval agent* that performs iterative search over the full document to collect all context-relevant evidence (see sec. 3.2). The second is an *LLM Answerer* that conditions on the retrieved chunks and generates the final response. Importantly, only the retrieval agent interacts with the original long context; the effective context length seen by the LLM Answerer depends solely on the retrieval hyperparameters

(e.g., number of retrieval steps $T$, maximum chunk length). Consequently, the time and memory complexity of the LLM Answerer with respect to the original context length $N$ are both $\mathcal{O}(1)$. The retrieval agent consists of two embedders: state embedder $E_s$ and action embedder $E_a$ (see sec. 3.2).

**Chunk Embedding.** The action embedder computes embeddings for chunks of the original document. If the document has length $N$ and the chunk size is $n_c$, embedding the entire document takes $\mathcal{O}\!\left(\frac{N}{n_c}\, t_{\text{act}}\right)$, where $t_{\text{act}}$ is the embedding time per chunk (treated as a constant). The action embedder performs a single pass over all chunks per retrieval episode; thus its complexity is linear in $N$, i.e., $\mathcal{O}(N)$.

**State Embedding.** The state embedder processes the state $K$ times per episode (once per search step). From the construction of the state (see fig. 1), the total cost over an episode is $\mathcal{O}(K\, t_{\text{state}})$, where state embedding time $t_{\text{state}}$ depends on $n_c$ and $K$, but not on $N$. Hence, the state embedder is $\mathcal{O}(1)$ with respect to document length $N$.

**Search Policy.** To select the next chunk at each step, we compute the inner product between the current state embedding and all action embeddings. With a naive implementation, selecting all $K$ actions over the episode requires $\mathcal{O}\!\left(K\, d_{\text{emb}}\, \frac{N}{n_c}\right) = \mathcal{O}(N)$, where $d_{\text{emb}}$ is the dimensionality of the embedding vectors. This can be reduced using approximate $k$NN methods that achieve sub-linear query time in practice (Malkov & Yashunin, 2018; Zhao et al., 2024).

**Overall time complexity.** Summing the terms above yields

$$\mathcal{O}\!\left(\frac{N}{n_c}\, t_{\text{act}} \; + \; K\, t_{\text{state}} \; + \; K\, d_{\text{emb}}\, \frac{N}{n_c}\right) \; = \; \mathcal{O}(N),$$

since $K$, $t_{\text{act}}$, $t_{\text{state}}$, and $d_{\text{emb}}$ do not depend on $N$.

**Space complexity.** The main part that directly depends on document length is the number of chunk embeddings we need to store: $\mathcal{O}\!\left(d_{\text{emb}}\, \frac{N}{n_c}\right) = O(N)$. In practice, embeddings are lightweight; GPU memory is mainly consumed by the LLM weights and the action embedder forward passes. By capping the action embedder's batch size (parameter `chunk_batch`), the growth of peak memory with $N$ becomes negligible.

**Training Time Efficiency.** A critical practical advantage of the Q-RAG framework is its efficient and rapid training convergence, as demonstrated in Figure 6. The learning curves depict the model's performance evolution on two distinct and challenging benchmarks: BabiLong QA2 and HotPotQA. The curves show a sharp initial rise in evaluation metric scores, followed by a stable plateau, indicating that the model quickly learns the core retrieval-augmented generation task. Notably, this convergence is achieved within approximately 12 hours of training time on a GPU setup.

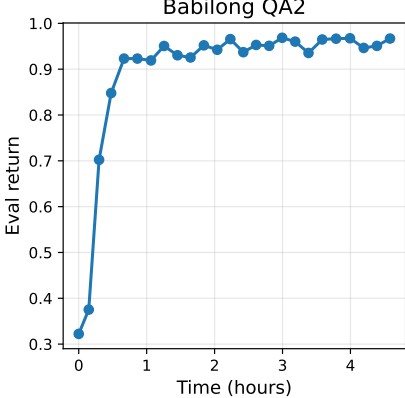 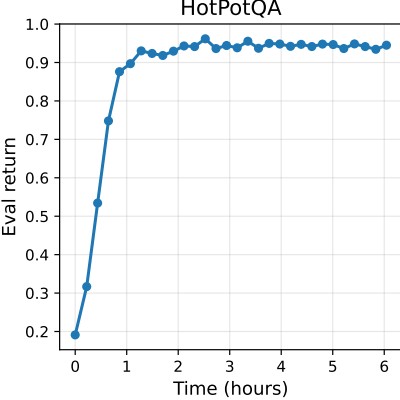

Figure 6: Learning curves for HotPotQA and BabiLong QA2 runs. Both graphs show the average episodic return with respect to training time.

# E    EXTRA QA RESULTS

Table 7 compares multi-step retrieval methods on HotPotQA-distractors, MuSiQue (in-distribution), and MuSiQue (out-of-distribution). It reports both fact-retrieval (Fact F1, Fact EM) and answer-generation (Ans F1, Ans EM) scores. Q-RAG and its planned variant (Plan Q-RAG) achieve strong overall results, especially on out-of-distribution data, while Beam-Retriever leads on HotPotQA but generalizes less robustly. Methods with missing entries did not report results for the corresponding dataset or metric.

Table 7: Comparison of methods on HotPotQA-distractors, MuSiQue (in-distribution), and MuSiQue (OOD). Bold text and underline denote the best and second best scores respectively.

| Methods | HotPotQA | | | | MuSiQue | | | | MuSiQue (OOD) | | | | Average | |
|---|---|---|---|---|---|---|---|---|---|---|---|---|---|---|
| | Fact F1 | Fact EM | Ans F1 | Ans EM | Fact F1 | Fact EM | Ans F1 | Ans EM | Fact F1 | Fact EM | Ans F1 | Ans EM | Ans F1 | Ans EM |
| Plan Q-RAG + QwQ-32B | 0.95 | 0.91 | 0.76 | 0.60 | 0.84 | **0.76** | **0.60** | **0.44** | 0.69 | 0.53 | 0.51 | 0.36 | **0.62** | **0.46** |
| Q-RAG+QwQ-32B | 0.93 | 0.89 | 0.76 | 0.59 | 0.81 | 0.72 | 0.59 | 0.43 | **0.71** | **0.55** | **0.52** | 0.37 | **0.62** | **0.46** |
| Beam Retriever+QwQ-32B | **0.97** | **0.94** | **0.77** | **0.61** | **0.86** | 0.69 | 0.59 | 0.43 | 0.61 | 0.36 | 0.40 | 0.27 | 0.59 | 0.44 |
| Search-R1 | 0.81 | 0.66 | 0.65 | 0.52 | – | – | – | – | **0.71** | **0.55** | 0.51 | **0.39** | – | – |
| Search-o1 | – | – | – | – | – | – | – | – | – | – | – | – | – | – |
| GraphReader | – | – | – | – | – | – | – | – | – | – | – | – | – | – |
| HippoRAG | – | – | – | – | – | – | – | – | – | – | – | – | – | – |

# F    TRAINING DETAILS

We trained the model with AdamW (learning rate $1.5 \times 10^{-5}$, $\beta_1$=0.9, $\beta_2$=0.98, $\epsilon$=$10^{-6}$, weight decay $5 \times 10^{-4}$). The learning rate followed a *linear* schedule: we used a warm-up of $1,000$ steps, then linearly decayed the rate to $10\%$ of its initial value over the remaining training steps. We applied gradient clipping with a maximum $\ell_2$ norm of $2.0$ and used gradient accumulation for $8$ steps. The base mini-batch size was $12$; with accumulation this yields an effective batch size of $12 \times 8 = 96$ per update (scaled by the number of devices if using distributed training).

In the objective and algorithmic components we set $\gamma$=0.99, $\alpha$=0.05, $\lambda$=0.5, and $\tau$=0.02. Action representations were capped at a maximum length of $220$ tokens.

The end-to-end training of a single model did not exceed 12 hours on a single `A100-80GB` GPU.

**Models per benchmark.** For open-domain QA benchmarks (*HotPotQA*, *MuSiQue*), we trained `multilingual-e5-large` and `Alibaba-NLP/gte-multilingual-base` encoders. For *RULER* and *BabiLong*, we trained `facebook/contriever`.

# G    EVALUATION DETAILS

**LLM Models for generation.** To compute answer-level metrics (Ans EM and Ans F1), we condition the QwQ-32B model on the question and the retrieved text chunks. All answer-generation results reported for Q-RAG and Plan Q-RAG on the HotPotQA and MuSiQue benchmarks were obtained under consistent generation settings: decoding with temperature $0.0$ and a maximum output length of `max_tokens = 8000`. For the BabiLong and RULER experiments, we instead used Qwen-4B with `max_tokens = 512` and reasoning disabled (`enable_thinking = False`).

**Retrieval configuration.** For Q-RAG, we limit the number of retrieval steps to $T = 2$ on HotPotQA; for RULER and BabiLong we use $T = 4$. The same step limits are used when evaluating Search-R1 and Beam Retriever.

We split documents into fixed-length, non-overlapping chunks, aiming not to break sentences across chunk boundaries. The chunk length is primarily determined by the context window of the embedders used in our main experiments (512 tokens) and the number of retrieval steps. For Needle-in-a-Haystack and BabiLong we use a chunk length of 64 tokens. For open-domain QA tasks we set the chunk length as a function of the number of retrieval steps i.e. for HotPotQA we segment the corpus into chunks of at most 220 tokens ($T = 2$); for MuSiQue we use action chunks of at most 110 tokens ($T = 4$). In additional experiments with a 'Alibaba-NLP/gte-multilingual-base' (8k context length) we use a chunk length of 256 tokens.

| Dataset | Setting | Chunk size | $T$ | Backbone retriever | Answering LLM |
|---------|---------|-----------|-----|-------------------|---------------|
| HotPotQA | Q-RAG / Plan Q-RAG | 220 | 2 | `multilingual-e5-large` | QwQ-32B |
| HotPotQA | Q-RAG (early stopping) | 256 | 5 | `Alibaba-NLP/gte-multilingual-base` | QwQ-32B |
| MuSiQue | Q-RAG / Plan Q-RAG | 110 | 4 | `multilingual-e5-large` | QwQ-32B |
| BabiLong | Q-RAG | 64 | 4 | `facebook/contriever` | Qwen3-4B |
| RULER | Q-RAG | 64 | 4 | `facebook/contriever` | Qwen3-4B |

Table 8: Retrieval and generation configuration for each dataset. Chunk size is in tokens; $T$ is the maximum number of retrieval steps.

**Fact-level metrics.** Let $S_{\mathrm{gt}}$ be the set of ground-truth supporting facts and $S_{\mathrm{pred}}$ be the set of predicted supporting facts returned by the retriever. Our Fact EM metric is defined as

$$\text{Fact-EM} = \begin{cases} 1, & \text{if } S_{\mathrm{gt}} \subseteq S_{\mathrm{pred}}, \\ 0, & \text{otherwise.} \end{cases}$$

Equivalently, in code: `em = 1.0 if gt_sf.issubset(pred_sf) else 0.0`. Thus Fact EM gives full credit whenever the prediction covers all ground-truth facts, even if it also contains additional, irrelevant chunks; it does not require the predicted and ground-truth sets to be exactly equal.

