# OpenReview forum: "Q-RAG: Long Context Multi‑Step Retrieval via Value‑Based Embedder Training"
_ICLR.cc/2026/Conference — ICLR 2026 Oral_

### Official Review · Reviewer_LyWb · 2025-10-24

**Soundness:** 2
**Presentation:** 2
**Contribution:** 2
**Rating:** 2
**Confidence:** 5

**Summary:**

This paper proposes an approach that fine-tunes an embedder model for multi-step retrieval through reinforcement learning. The proposed method, Q-RAG, achieves state-of-the-art results on long-context benchmarks including Babilong and Needle-in-a-haystack tasks.

**Strengths:**

- By decoupling the retriever from the LLM and training only a lightweight embedder, the proposed approach is practical, compared to other approaches that require fine-tuning the entire LLMs.
- This paper introduces a relative positional encoding scheme for context embeddings. This allows the agent to reason about the temporal relationships between chunks.
- The experiments demonstrate scalability and robustness, achieving state-of-the-art results on benchmarks with contexts up to 10 M tokens.

**Weaknesses:**

- The paper’s analysis of the multi-step retrieval process is under-explored. While the method relies on a maximum step budget, T, this critical hyperparameter is not specified for the experiments, hindering reproducibility. Furthermore, the paper lacks an investigation into the agent’s learned stopping behavior, i.e., it is unclear if the agent effectively utilizes the STOP action to terminate retrieval early. A sensitivity analysis on how performance varies with different values of T is also absent. This omission is significant, as the optimal number of retrieval steps is often unknown in practice, leaving the method’s robustness and practical applicability in question.
- The paper’s presentation could be significantly improved. Several key results and concepts are not properly introduced or referenced, which disrupts the reader’s flow. For instance, the “Plan-Q-RAG” varian appears abruptly in Table 2 without prior introduction or motivation in the main body of the paper. Later, I found its explanation in the appendix. Additionally, Figure 2 is not explicitly mentioned anywhere in the text.
- The empirical validation for the proposed temporal reasoning mechanism appears narrow. The authors claim this mechanism is a major technical contribution, yet its effectiveness is demonstrated exclusively on the Babilong benchmark. To substantiate the generalizability and impact of this technique, the evaluation should have included a broader range of tasks where temporal or sequential understanding is paramount. For example, testing on datasets that require reasoning over chronologically ordered documents or procedural texts (e.g., [1]) would provide more robust evidence and strengthen the authors’ claims.

[1] Karpinska, Marzena, et al. "One Thousand and One Pairs: A “novel” challenge for long-context language models." Proceedings of the 2024 Conference on Empirical Methods in Natural Language Processing. 2024.

**Questions:**

Can you explain all the points I raised in the Weaknesses section?

---

> ### Author Response · Authors · 2025-12-03
> **Rebbuttal to Reviewer LyWb [part 1]**
>
> We thank the reviewer for the helpful comments. We address the weaknesses and questions below:
>
> **[W1] Retrieval step budget and Early stopping policy**
>
> Some technical details were omitted from the main text because we provided the code and assumed it would be available to reviewers at the time of reviewing. In the open-domain QA experiments, following Beam Retriever [1], we set $T$ to the maximum number of support facts over all samples in the benchmark: $T = 2$ for HotpotQA and $T = 4$ for Musique. In all other experiments we used $T = 4$. For the reproduced retrieval baselines, we also limited the number of retrieval steps to the same $T$ for a fair comparison. We will add a detailed description of the inference hyperparameters in the appendix "Evaluation Details".
>
> We did not use the stopping action in the original experiments. However, to address your questions we conducted new experiments to study a simple cost-free stopping policy for the Q-RAG agent. For this, we modified training: instead of ending an episode when all support facts are found, we fixed the episode length to $T=5$ steps for HotpotQA (chunk length = 256 tokens) and $T=6$ steps for Babilong (chunk_length = 64 tokens). The agent receives a reward only at the step when all support facts are collected, and 0 at all later steps. In this setup, the true value of the $Q^{\pi}$ function for any policy is 0 once all necessary information has been found, so the learned $Q$ value stays high before that point and drops toward 0 afterward. For both experiments we used the `Alibaba-NLP/gte-multilingual-base` as backbone (with 8k context length) model.
>
> This lets us define a stopping criterion based on a Q-value threshold: the agent stops when $Q$-value  falls below a chosen threshold. Results for HotpotQA and Babilong are shown in the tables below.  We will also include plots and a detailed description in the appendix section "Early Stopping Experiments".
>
> **Table 1. HotPotQA early stopping experiments**
>
> | Q threshold | stopped early | stopped late | perfect stop | Episode len | Fact EM | Fact F1 | Answer EM | Answer F1 |
> |-------------------|--------------:|--------------:|-------------:|------------:|--------:|--------:|-------:|-------:|
> | 0.0               | 0.015         | 0.397         | 0.588        | 2.83        | 0.954   | 0.842   | 0.592  | 0.761  |
> | 0.1               | 0.061         | 0.061         | 0.878        | 2.23        | 0.909   | 0.915   | 0.593  | 0.756  |
> | 0.2               | 0.088         | 0.020         | 0.891        | 2.13        | 0.883   | 0.916   | 0.587  | 0.752  |
> | 0.3               | 0.105         | 0.006         | 0.889        | 2.08        | 0.867   | 0.914   | 0.585  | 0.747  |
> | 0.4               | 0.119         | 0.002         | 0.879        | 2.05        | 0.853   | 0.911   | --     | --     |
> | 0.5               | 0.133         | 0             | 0.866        | 2.02        | 0.839   | 0.906   | --     | --     |
> | 0.6               | 0.145         | 0             | 0.855        | 2.00        | 0.828   | 0.902   | --     | --     |
> | 0.7               | 0.158         | 0             | 0.842        | 1.96        | 0.815   | 0.894   | --     | --     |
> | 0.8               | 0.203         | 0             | 0.797        | 1.82        | 0.771   | 0.845   | --     | --     |
> | 0.9               | 0.419         | 0             | 0.581        | 1.27        | 0.562   | 0.617   | --     | --     |
> | 1.0               | 0.911         | 0             | 0.089        | 0.21        | 0.086   | 0.109   | --     | --     |
> | 1.1               | 1.000         | 0             | 0            | 0.00        | 0       | 0       | --     | --     |

---

> ### Author Response · Authors · 2025-12-03
> **Rebbuttal to Reviewer LyWb [part 2]**
>
> **Table 2. Babilong QA2 early stopping experiments**
>
> | Q threshold       | stopped early | stopped later | perfect stop | Episode len | Fact EM | Fact F1 | Ans EM | Ans F1 |
> |-------------------|--------------:|--------------:|-------------:|------------:|--------:|--------:|-------:|-------:|
> | 0.00              | 0.000         | 0.994         | 0.006        | 6.00        | 0.996   | 0.499   | 0.884  | 0.884  |
> | 0.10              | 0.000         | 0.498         | 0.502        | 2.86        | 0.996   | 0.845   | 0.944  | 0.944  |
> | 0.20              | 0.000         | 0.036         | 0.964        | 2.29        | 0.996   | 0.949   | 0.976  | 0.976  |
> | 0.30              | 0.006         | 0.010         | 0.984        | 2.25        | 0.990   | 0.952   | 0.970  | 0.970  |
> | 0.40              | 0.008         | 0.002         | 0.990        | 2.24        | 0.988   | 0.953   | 0.970  | 0.970  |
> | 0.50              | 0.008         | 0.000         | 0.992        | 2.23        | 0.988   | 0.954   | 0.972  | 0.972  |
> | 0.60              | 0.016         | 0.000         | 0.984        | 2.21        | 0.980   | 0.948   | 0.968  | 0.968  |
> | 0.70              | 0.042         | 0.000         | 0.958        | 2.16        | 0.954   | 0.934   | 0.948  | 0.948  |
> | 0.80              | 0.112         | 0.000         | 0.888        | 2.06        | 0.884   | 0.905   | 0.884  | 0.884  |
> | 0.90              | 0.177         | 0.000         | 0.823        | 1.92        | 0.820   | 0.861   | 0.830  | 0.830  |
> | 1.00              | 0.930         | 0.000         | 0.070        | 0.22        | 0.070   | 0.107   | 0.230  | 0.230  |
>
> Fact EM, Fact F1, Answer EM and Answer F1 measure the quality of the Q-RAG agent and the end-to-end QA system. In particular, Fact EM is the fraction of trajectories where all supporting facts are found, while Fact F1 also reflects the proportion of noisy facts among the retrieved ones. The columns “stopped_early”, “perfect_stop” and “stopped_later” evaluate the stopping criterion based on the Q-value. They are computed only over successful trajectories, where the ideal stopping time $t_{oracle}$ is the step when the last supporting fact is retrieved. Each of these three columns reports the fraction of cases where the Q-threshold criterion fires earlier than $t_{oracle}$, exactly at $t_{oracle}$, or after $t_{oracle}$ respectively. On the Babilong task, using the automatic stopping criterion yields almost perfect results for $q_{\text{threshold}}$ between 0.3 and 0.6. On HotpotQA, for $q_{\text{threshold}}$ values between 0.1 and 0.3 performance with this stopping criterion matches the performance achieved with fixed $T$ (see Table 2 in the paper).
> We also note that another natural option is to let the LLM itself decide when to stop retrieval. Smaller open models of around 32B parameters may struggle with this, but best current models could generate a strong stopping policy.
>
> **[W2] Presentation clarity: Plan-Q-RAG and Figure 2**
>
> We thank the reviewer for pointing this out. In the revised version, we will explicitly introduce Plan-Q-RAG in the main text and carefully check that all figures, tables and concepts are properly introduced and referenced in the main body of the paper.
>
> **[W3] Extra benchmarks to expand experiments on temporal encoding**
>
> The main contribution of Q-RAG is a value-based reinforcement learning method that fine-tunes the embedder for efficient multi-step retrieval in latent space. This is cheaper than fine-tuning the LLM and allows our retrieval agent to be paired with any LLM. Most of our experiments do not rely on the temporal encoding component.
>
> The suggested NoCha benchmark [2] does not match our training and evaluation setup, as it targets claim verification over full books. Although the Q-RAG training algorithm can be applied to other multi-step retrieval tasks, in this work we did not collect enough data to train a universal embedder with temporal positional encoding over chunks. Therefore, applying Q-RAG to temporally sensitive claim-verification would require suitable training data, which NoCha does not provide.
>
> We agree that a broader evaluation of the temporal encoding would further support its usefulness, but time and resource limits in the rebuttal phase prevent us from adding results on another temporal reasoning dataset. We believe the current results already support our main claims about efficient multi-step retrieval and state-of-the-art performance on long-context benchmarks.
>
> References:
> 1. Zhang, Jiahao, et al. "End-to-end beam retrieval for multi-hop question answering." Proceedings of the 2024 Conference of the North American Chapter of the Association for Computational Linguistics: Human Language Technologies (Volume 1: Long Papers). 2024.
> 2. Karpinska, Marzena, et al. "One Thousand and One Pairs: A “novel” challenge for long-context language models." Proceedings of the 2024 Conference on Empirical Methods in Natural Language Processing. 2024.

---

### Official Review · Reviewer_nFRL · 2025-10-31

**Soundness:** 4
**Presentation:** 3
**Contribution:** 3
**Rating:** 8
**Confidence:** 4

**Summary:**

The paper presents Q-RAG, a framework that formulates multi-step retrieval in retrieval-augmented generation (RAG) as a reinforcement learning problem. Two transformer encoders are trained: a state encoder that represents the current retrieval context (the initial query and previously retrieved chunks), and an action encoder that represents each candidate document chunk. The similarity between the two embeddings defines a Q-value, which indicates the estimated utility of retrieving that chunk in the current context. Training uses Soft Q-Learning with λ-return, where the model receives a binary outcome-based reward depending on whether all supporting facts are retrieved. During inference, the model iteratively embeds the current state, scores candidate chunks by their Q-values, selects the most relevant one, and updates the state until retrieval is complete. Experiments are conducted on HotpotQA, MuSiQue, RULER (1M tokens), and BabiLong (10M tokens) to evaluate retrieval accuracy and computational efficiency.

**Strengths:**

- The paper presents a clear formulation of multi-step retrieval as a reinforcement learning process, replacing traditional query rewriting with a learned retrieval policy. This modeling perspective provides a coherent computational framework for retrieval-augmented generation.
- The proposed model adopts a dual-encoder design, where a state encoder and an action encoder jointly estimate Q-values to guide multi-step retrieval decisions. Both encoders are compact transformer models, showing that a small embedding-based retriever can learn retrieval behavior previously handled by large language models.
- The training framework integrates Soft Q-Learning with λ-return, which helps stabilize optimization under sparse outcome rewards. The method shows that an embedding model can reach retrieval accuracy comparable to reinforcement-learning–finetuned large language model retrievers while operating at roughly 10–20× lower computational cost.
- The experimental evaluation covers multiple datasets, including HotpotQA, MuSiQue, and synthetic long-context settings (RULER, BabiLong), demonstrating consistent retrieval performance and scalability to contexts up to 1M–10M tokens.

**Weaknesses:**

- The model’s architecture and training setup restrict its ability to directly handle long or unstructured contexts. Both encoders have short input windows, and the approach depends on datasets with annotated supporting facts, which limits generalization to open-domain or unlabeled corpora.
- The paper lacks theoretical analysis of the proposed Q-learning framework. No formal discussion of convergence, optimality, or error bounds is provided, and the training stability is supported only by empirical observations.
- The reward design relies on a single outcome-based binary signal. While λ-return mitigates the sparsity issue, the paper does not discuss other possible reward shaping or exploration strategies.

**Questions:**

- Do the authors observe any evidence of policy collapse or repeated retrieval of the same chunks during training?
- How well does the learned policy generalize to unseen domains or corpora without retraining?

---

> ### Author Response · Authors · 2025-12-03
> **Answer for Reviewer nFRL [part 1]**
>
> We thank the reviewer for the insightful feedback and valuable comments!
>
> **W1: The model’s architecture and training setup restrict its ability to directly handle long or unstructured contexts.** Both points highlight different facets of Q-RAG's design philosophy.
>
> Regarding Embedder Context Window: The short context window is a property of the particular encoder backbone, not a restriction of Q-RAG itself. In the revised version we add experiments with a longer-context encoder, Alibaba-NLP/gte-multilingual-base with an 8k token window (see Appendix, Early Stopping Experiments). This model is slightly smaller than E5, yet by simply swapping the backbone in the config we increase the maximum state size of the retrieval agent by about 16x. The experiments show that, when combined with early stopping based on the learned Q-function, this encoder achieves performance comparable to E5 on HotPotQA and BabiLong. In principle Q-RAG can use even larger encoders with longer context windows. The main limitation is hardware. For example, encoders with more than one billion parameters would require more than a single A100 GPU. Additionally, to compensate for the loss of global positional information from chunking, we introduce a novel relative positional encoding $\rho_t(i)$ (Eq. 6–7). This encoding explicitly informs the action embedder about a chunk's position relative to the already retrieved facts, enabling temporal reasoning without a global view. We also demonstrate that Q-RAG can exploit this relative positional signal on BabiLong tasks, many of which require positional and temporal reasoning about the relative order of facts (for example, retrieving the last supporting fact or a fact that occurred before a given fact).
>
>
> Regarding Support-Fact Supervision: This is a valid limitation of our current experiments. However, the RL framework is reward-agnostic. We used support facts for a clean evaluation, but the system can be trained with LLM-based rewards (using final answer quality) or self-supervised signals, making it applicable to unlabeled corpora. We mention in the conclusion that 'using structured LLM feedback as a reward signal' is a promising direction, and this critique reinforces its importance.
>
> **W2: The paper lacks theoretical analysis of the proposed Q-learning framework.**
>
> Our Q-RAG algorithm is built on a solid, well-defined Finite-Horizon Markov Decision Process (MDP). The theoretical guarantees we can appeal to are as follows:
>
> For the Maximum Entropy RL Framework: Our use of soft Q-learning follows the formulation of [Haarnoja et al., 2018, Soft Actor-Critic]. This framework has known convergence properties. Specifically, for a given policy, the soft Q-value can be shown to converge to the solution of the soft Bellman equation under standard conditions.
>
> Regarding PQN: PQN's key innovation is removing the replay buffer by using a projected Bellman operator, which has theoretical grounding. Its convergence relies on the contractive property of this projected operator. Our departure—introducing soft value functions and target networks—was an empirical choice for stability.
>
> We want to point out that Appendix A contains a universal approximation result for our Q-function parameterization. We prove that representing Q as the inner product of state and action embeddings can approximate any continuous function arbitrarily well (see Appendix A).

---

> > ### Author Response · Authors · 2025-12-04
> > **Answer for Reviewer nFRL [part 2]**
> >
> > **W3: The reward design relies on a single outcome-based binary signal.**
> >
> > Thank you for this point. You are correct that our reward is sparse. However, with our typical episode length of $T < 10$ steps (as required by multi-hop QA benchmarks), the credit assignment problem is substantially mitigated. The $\lambda$-return effectively propagates the terminal reward back through this short trajectory. In such short-horizon MDPs, complex reward shaping is often unnecessary and can even be harmful by introducing bias. Our choice of a sparse, outcome-based reward follows the principle of using the simplest sufficient signal. It directly optimizes the true objective—retrieving all support facts—without potential distortions from heuristic intermediate rewards.
> >
> > **[Q1] Do the authors observe any evidence of policy collapse or repeated retrieval of the same chunks during training?**
> >
> > We did monitor for these failure modes, and our design includes specific mechanisms to prevent them.
> >
> > 1. via Action Space Masking: The most fundamental safeguard is that our MDP formulation explicitly defines the action space at step t as $A_t = C \setminus (a_0, ..., a_{t-1})$. Once a chunk is retrieved, it is removed from the available action set for the remainder of the episode.
> >
> > 2. via Maximum Entropy RL: Our use of a maximum-entropy (soft) Q-learning objective with an annealed temperature $\alpha$ is specifically designed to combat policy collapse. The entropy term in the loss function actively encourages exploration and penalizes the policy for becoming too deterministic too quickly. The ablation study (Table 3, "Q-RAG w.o.Soft-Q") shows a performance drop when this term is removed, which is empirical evidence that the entropy regularization is crucial for stability and preventing premature convergence to a suboptimal policy.
> >
> > **[Q2] How well does the learned policy generalize to unseen domains or corpora without retraining?**
> >
> > Q-RAG demonstrates impressive generalization along the axis of context length and modest generalization across related QA tasks. Table 2 shows results on the Musique dataset, which was used for out-of-distribution (OOD) evaluation after training on HotPotQA. Here, Q-RAG performs competitively, matching or outperforming other fine-tuned methods (BeamRetriever, Search-R1, RAG-RL).

---

### Official Review · Reviewer_sUPb · 2025-11-01

**Soundness:** 3
**Presentation:** 4
**Contribution:** 3
**Rating:** 8
**Confidence:** 4

**Summary:**

This paper introduces Q-RAG, a novel and resource-efficient framework for multi-step retrieval in Retrieval-Augmented Generation (RAG) systems. The authors' primary contribution is a paradigm shift away from fine-tuning the Large Language Model (LLM) itself, focusing instead on training a smaller, more efficient embedder model using value-based reinforcement learning (RL). The paper claims three main contributions: (1) a new method for training a multi-step retrieval agent using temporal difference RL, which is exceptionally resource-efficient (trainable on a single GPU); (2) state-of-the-art performance on ultra-long context benchmarks (up to 10M tokens) like Babilong and RULER; and (3) a novel relative positional encoding mechanism that enables temporal reasoning during the retrieval process. This approach is presented as a modular, flexible, and scalable solution to the challenges of multi-hop reasoning and "needle-in-a-haystack" problems in extremely long documents.

**Strengths:**

- **Novelty and Technical Soundness:** The core idea of fine-tuning the embedder instead of the LLM for multi-step retrieval is a significant and elegant contribution. The formulation of the problem as a Markov Decision Process (MDP) is well-defined, and the factorization of the Q-function into an inner product of state and action embeddings is a clever design choice, theoretically justified in the appendix. The introduction of dynamic relative positional encoding to handle temporal reasoning is particularly innovative and is empirically shown to be highly effective on the Babilong QA3 task.

- **Exceptional Resource Efficiency and Accessibility:** The ability to train the entire system on a single A100 GPU is a major practical advantage. This dramatically lowers the barrier to entry for research in multi-step RAG, making it accessible to a much wider community compared to LLM-tuning methods that require large GPU clusters.

- **Modularity and Flexibility:** By decoupling the retriever from the generator, the trained Q-RAG embedder can be paired with any LLM, including powerful proprietary models like GPT-4. This "plug-and-play" nature is a significant strength in the current AI ecosystem.

- **Strong Empirical Results and Scalability:** The paper demonstrates state-of-the-art or highly competitive performance across a range of challenging benchmarks, including ultra-long context (Babilong, RULER) and standard multi-hop QA (HotpotQA, Musique). The model's ability to maintain high performance as context length scales to 10 million tokens is particularly impressive and clearly demonstrates the scalability of the proposed method.

- **Reproducibility:** The authors provide detailed experimental setups and ablation studies justifying their architectural choices (e.g., using soft Q-learning and target networks). They also state their intention to release code, which strengthens the paper's contribution.

**Weaknesses:**

- **Critical Dependence on Supervised Reward Signal:** The paper's most significant weakness is its reliance on a sparse terminal reward based on the retrieval of a pre-defined set of "ground-truth support facts ($F^*$)." While this provides a stable training signal, it severely limits the method's real-world applicability. Such meticulously labeled datasets are extremely rare and prohibitively expensive to create for new domains (e.g., legal, medical, enterprise knowledge). This dependency relegates the current method to a powerful tool for academic benchmarks rather than a general-purpose, deployable solution.

- **Underestimation of the LLM-as-a-Judge Problem:** The paper acknowledges the limitation of supervised rewards and briefly suggests using an LLM to generate a reward signal as "future work." This suggestion dangerously overlooks the well-documented and critical failure modes of the "LLM-as-a-Judge" paradigm. Recent research has extensively shown that LLM judges are systematically vulnerable to "reward hacking" and "master key" attacks, where superficial, semantically meaningless inputs (e.g., punctuation, phrases like "Thought process:") can elicit false positive rewards. An RL agent like Q-RAG would rapidly learn to exploit these vulnerabilities, leading to a collapsed policy that generates "master keys" instead of performing meaningful retrieval. The paper's failure to address this critical challenge makes its proposed path to unsupervised application seem naive and unviable in its current form.

- **Limited Compositional Reasoning in the Retriever:** While the relative positional encoding enables temporal reasoning, the retriever itself does not perform more complex compositional reasoning, such as comparison or calculation. The task of synthesizing information and drawing a final conclusion is entirely offloaded to the final, frozen LLM. The system's performance is therefore contingent on the capabilities of this final LLM, and it may fail on questions that require intricate reasoning over the retrieved facts.

- **Isolation from the Broader Advanced RAG Ecosystem:** The proposed method operates in isolation from other well-established advanced RAG techniques like hybrid search (combining dense and sparse retrieval), parent-child retrieval, or sophisticated reranking modules. Integrating Q-RAG's RL-based retriever into a more comprehensive, production-style pipeline could be beneficial, but this is not explored.

**Questions:**

- Given that creating datasets with ground-truth support facts is infeasible for most real-world applications, what is your proposed concrete path toward adapting Q-RAG to new domains where such supervision is unavailable?

- Your paper suggests using LLM-based rewards as a future direction. How do you plan to address the now well-established problem of "reward hacking," where RL agents learn to generate superficial "master keys" to fool LLM judges? Without a robust defense against this, wouldn't the training process collapse?

- Could you elaborate on the model's limitations in tasks requiring explicit compositional reasoning (e.g., "Which of these two products has a higher rating?" which requires retrieval of two facts and a comparison)? How does Q-RAG's performance depend on the reasoning capability of the final, frozen LLM used for generation?

- Have you considered the performance implications of integrating the Q-RAG retriever as the first stage in a more complex pipeline that includes, for example, a cross-encoder reranker? Would the greedy, step-by-step retrieval policy still be optimal in such a setting?

---

> ### Author Response · Authors · 2025-12-04
> **Rebbuttal to Reviewer sUPb [part 1]**
>
> We thank the reviewer for the careful reading of our paper and for the detailed and insightful comments!
>
> **[Q1/W1] Dealing with lack of ground truth support facts.**
>
> The first path is to rely on synthetic data. While human annotation of support facts is expensive, there is already a substantial line of work where LLMs automatically generate training signals for retrievers and RAG systems. For example, Syntriever [1] generates relevant and hard negative passages and trains a dense retriever purely on LLM synthesized data. SWIM-IR [2] and InPars [3] use LLMs to generate queries for existing documents and show that such synthetic query-passage pairs can successfully replace costly manual labels for dense retrieval. In applied domains such as chemistry, ChemLit-QA [4] constructs synthetic question–answer–context triples from domain literature and uses them to train and evaluate RAG systems. In the same spirit, for new domains we can either generate candidate retrieved chunks (including negatives) or generate question–answer pairs for existing documents, which provides an approximate analogue of our support facts without requiring full manual supervision.
>
> The second path is to use LLM feedback to assess the quality of the retriever. We mentioned LLM based rewards as a potential extension, but this does not have to appear in a form of “LLM as a judge” reward. An LLM can simply generate candidate answers given the retrieved context, and we produce reward by comparing predictions to ground truth answers using other metrics (e.g., EM, F1 score, uncertainy measures). We discuss the feasibility of using LLM feedback in more detail in our answer to Question 2.
>
> **[Q2/W2] On Rewards from LLM feedback and reward hacking**
>
> We acknowledge that the reviewer’s concerns about using LLM feedback and the risk of reward hacking point to a genuine problem that requires careful and cautious treatment. In a sense, this reinforces our decision not to treat using LLM feedback as another experiment in this paper, but rather to leave it as a dedicated future research direction that requires focused attention.
>
> Regarding the concern about the agent discovering master-key tokens and engaging in reward hacking [5], our reading of the existing literature suggests that these failures typically arise in settings where the agent itself is an LLM that directly generates the token sequences fed to the LLM judge. Our proposed use of LLM feedback differs in an important way. In our setting, however, the LLM and its prompt template are fixed, and the RL agent never directly generates input for an LLM-judge. The agent operates only in the space of chunk embeddings, and each action corresponds to selecting an existing chunk from a fixed corpus. This strongly constrains the space of possible adversarial inputs. To perform reward hacking, the agent would need to repeatedly select chunks that already contain effective “master key” patterns, which is unlikely to happen.
>
> At the same time, there is a growing body of evidence that carefully designed LLM based supervision can be successfully used to generate a training signal. REPLUG [6] uses LLM log likelihoods to train a single-step retriever and shows consistent gains without human labels. LUF uses combined LLM and user feedback in a contrastive objective to improve RAG under domain shift [7], and ReG leverages LLM preferences to align weak graph based retrievers and achieves strong performance on graph RAG benchmarks [8]. In parallel, new methods for robust reward modeling explicitly target reward hacking [9]. Together, these results suggest that LLM feedback can be a practical training signal when combined with appropriate constraints and robust reward modeling. Our setting inherits some of these advantages due to the restricted action space, and we view designing and validating a robust LLM feedback pipeline for Q-RAG as a natural direction for follow up work rather than something that can be fully resolved within this paper.
>
> [1] Kim, M., Baek, S. “Syntriever: How to Train Your Retriever with Synthetic Data from LLMs.” Findings of NAACL 2025.
>
> [2] Thakur, N. et al. “Leveraging LLMs for Synthesizing Training Data Across Languages” (SWIM-IR). NAACL 2024.
>
> [3] Bonifacio, D. et al. “InPars: Data Augmentation for Information Retrieval using Large Language Models.” SIGIR 2022.
>
> [4] Wellawatte, G. et al. “ChemLit-QA: A human evaluated dataset for chemistry RAG tasks.” NeurIPS ML4PS 2024.
>
> [5] Zhao et al. “One Token to Fool LLM-as-a-Judge.” 2025.
>
> [6] Shi et al. “REPLUG: Retrieval-Augmented Black-Box Language Models.” 2023.
>
> [7] “LLM and User Feedback Based Contrastive Learning Improves Retrieval-Augmented Generation When Question and Answer Domains Shift (LUF).” 2025.
>
> [8] Zou et al. “Weak-to-Strong GraphRAG: Aligning Weak Retrievers with Large Language Models for Graph-based Retrieval Augmented Generation.” 2025.
>
> [9] Liu et al. “RRM: Robust Reward Model Training Mitigates Reward Hacking.” 2024.

---

> > ### Author Response · Authors · 2025-12-04
> > **Rebbuttal to Reviewer sUPb [part 2]**
> >
> > **[Q3/W3] On compositional reasoning in the retriever and the role of the frozen LLM**
> >
> > Our retriever is itself a Transformer model, so in principle it can express all computations that are needed for complex reasoning over the facts encoded in its current state. Its main limitation is that it is not a generative model, therefore it cannot increase the amount of computation by producing long chains of intermediate tokens. Within a single decision, however, it can compute complex relations between already retrieved facts and use them to form the latent query embedding that drives the next retrieval step.
> >
> > In our experiments we observe that the retriever’s accuracy in finding correct support facts is higher than, or comparable to, the end to end QA accuracy of the system (see table below). This suggests that the main bottleneck is the frozen LLM used for answer generation, not the retriever ability to find relevant information.
> >
> > **Difference between retriever ability to find all support facts and final QA accuracy of the whole pipeline:**
> >
> > | Dataset                         | Fact EM | Ans EM | Fact - Ans Diff |
> > |---------------------------------|:-------:|:------:|:----------------|
> > | HotPotQA                        |  0.89   |  0.59  |      0.3        |
> > | Musique                         |  0.72   |  0.43  |      0.29       |
> > | Musique (OOD)                   |  0.55   |  0.37  |      0.18       |
> > | Babilong [128k]                 | 0.96-1. | 0.89-1.|     0.-0.09     |
> > | RULLER Niah tasks [128k]        |  1.     |  1.    |      0.         |
> >
> >
> > This gap is likely due to the fact that in this work we used relatively small frozen LLMs (Qwen-4B, Qwen-8B, QwQ-32B). At the same time, the Q-RAG architecture can be combined with much stronger proprietary or open models. It is therefore reasonable to expect that using more powerful generators would reduce or eliminate the current gap between retrieval quality and final answer quality.
> >
> > **[Q4/W4] On complex RAG pipelines**
> >
> > Q-RAG training does not require a greedy search policy or prevent integration with more complex retrieval pipelines. In Table 2 we evaluate a Plan-Q-RAG variant that uses beam search to choose a sequence of retrieval actions (the method is described in Appendix B). This shows that Q-RAG can be combined with more advanced planning that explores several retrieval branches in parallel. In our experiments, we used the Q-function value to score trajectories and did not observe clear gains over the greedy strategy. However, with a stronger scoring function or a dedicated evaluation model, such a pipeline could outperform greedy retrieval.
> >
> > Regarding the broader point about using Q-RAG in a more complex RAG pipeline with rerankers, hybrid search, or multiple retrievers, we agree that such components could further improve final system quality. But we want to stress, that our work focuses on the training method for multi-step retrieval, and we view these kinds of pipeline-level modifications as orthogonal to our main scope, precisely because complex RAG pipelines are typically agnostic to how candidate chunks are obtained and can be combined with any underlying retriever, including Q-RAG. Adding these components would also make fair comparison with existing baselines much harder. In particular, majority of the RAG focused baselines in our paper use hierarchical retrieval or sophisticated reranking, although they could also benefit from such components. Including these techniques for our agent would make the comparison unfair.

---

### Official Review · Reviewer_rFyU · 2025-11-02

**Soundness:** 3
**Presentation:** 3
**Contribution:** 3
**Rating:** 6
**Confidence:** 3

**Summary:**

Q-RAG formulates multi-step retrieval as a MDP process where the state is the embedding representation of the query and past retrieved chunks, the action is the embedding of the candidate next chunk to retrieve, and Q is defined as the dot product of the state and the action embedding. Reward is defined as whether all ground truth chunks are retrieved and are propagated back via TD to train the state and action embedders. The main contribution of Q-RAG is to train a multi-hop retriever using temporal-difference RL. The authors also claim Q-RAG to be more efficient in training and inference, as only the retriever is trained and iteratively used during inference

**Strengths:**

1. The method is intuitively sound and novel, with theoretical justification of using dot product as an approximation for the Q-function in the appendix.
2. The experimental setup is comprehensive with good coverage on multiple tasks requiring multi-step retrieval, spanning commonsense reasoning, NIAH, and multi-hop retrieval.
3. Numerous baselines are included for different datasets, and the baselines are very recent.
4. Good ablation study to demonstrate the necessity of various key design choices in Q-RAG.

**Weaknesses:**

1. In all experiments, the choice of the total number of retrieval steps T is very important because you need to know when to stop retrieving and have the reader LLM answer the question. However, the choice of T is not discussed at all.
2. The authors claimed efficiency in training and inference. Though it is intuitively true, the discussion regarding training efficiency was insufficient and was not studied. Only the inference runtime was partially studied in Figure 3c.
3. The experimental setups are pretty vague. What off-the-shelf models were used as the state embedder and action embedder? What reader LLM are used for different baselines for different datasets? Why the different choices? The paper can also benefit from brief introductions to each dataset and elaborating on what chunking strategy was used for each dataset for readers not familiar with them.
4. Presentation: Figure 3c’s bottom has been cut off and typo “Babylon” on line 42

**Questions:**

see the weakness point 3

---

> ### Author Response · Authors · 2025-12-03
> **Answer to Reviewer rFyU**
>
> Thank you for your thoughtful feedback! We have carefully revised the manuscript to address each of the points raised.
>
> **[W1]  Total number of retrieval steps T **
>
> Thank you for raising this point. Some technical details were omitted from the main text because we provided the code and assumed it would be available to reviewers at the time of reviewing. In the open-domain QA experiments, following Beam Retriever [1], we set $T$ to the maximum number of support facts over all samples in the benchmark: $T = 2$ for HotpotQA and $T = 4$ for Musique. In all other experiments we used $T = 4$. For the reproduced retrieval baselines, we also limited the number of retrieval steps to the same $T$ for a fair comparison. We added a detailed description of the inference hyperparameters in the appendix "Evaluation Details" in the revisited version. We included addtional experiments realated to the number of retrieval steps.
>
>
> We have added new experiments concerning methods for automatically stopping retrieval based on the value of the Q-function at the current step. A detailed description of these experiments is provided below. Additional figures and tables can be found in the paper in the "Early Stopping Experiments" section of the appendix.
>
> To obtain Q-function estimates that can be used to perform early stopping we first modified the training of the Q-RAG agent as follows: instead of ending an episode when all support facts are found, we fixed the episode length to $T=5$ steps for HotpotQA (chunk length = 256 tokens) and $T=6$ steps for Babilong (chunk_length = 64 tokens). The agent receives a reward only at the step when all support facts are collected, and 0 at all later steps. In this setup, the true value of the $Q^{\pi}$ function for any policy is 0 once all necessary information has been found, so the learned $Q$ value stays high before that point and drops toward 0 afterward. For both experiments we used the `Alibaba-NLP/gte-multilingual-base` as backbone model. This model differs from the backbone model we used for the main experiments, as it is smaller but has a longer context length (8k tokens), which allows us to increase the maximum number of retrieval steps without sacrificing chunk length.
>
> This lets us define a stopping criterion based on a Q-value threshold: the agent stops when $Q$-value  falls below a chosen threshold. Results for HotpotQA and Babilong are shown in the tables below.
>
> **Table 1. HotPotQA early stopping experiments**
>
> | Q threshold | stopped early | stopped late | perfect stop | Episode len | Fact EM | Fact F1 | Answer EM | Answer F1 |
> |-------------------|--------------:|--------------:|-------------:|------------:|--------:|--------:|-------:|-------:|
> | 0.0               | 0.015         | 0.397         | 0.588        | 2.83        | 0.954   | 0.842   | 0.592  | 0.761  |
> | 0.1               | 0.061         | 0.061         | 0.878        | 2.23        | 0.909   | 0.915   | 0.593  | 0.756  |
> | 0.2               | 0.088         | 0.020         | 0.891        | 2.13        | 0.883   | 0.916   | 0.587  | 0.752  |
> | 0.3               | 0.105         | 0.006         | 0.889        | 2.08        | 0.867   | 0.914   | 0.585  | 0.747  |
> | 0.4               | 0.119         | 0.002         | 0.879        | 2.05        | 0.853   | 0.911   | --     | --     |
> | 0.5               | 0.133         | 0             | 0.866        | 2.02        | 0.839   | 0.906   | --     | --     |
> | 0.6               | 0.145         | 0             | 0.855        | 2.00        | 0.828   | 0.902   | --     | --     |
> | 0.7               | 0.158         | 0             | 0.842        | 1.96        | 0.815   | 0.894   | --     | --     |
> | 0.8               | 0.203         | 0             | 0.797        | 1.82        | 0.771   | 0.845   | --     | --     |
> | 0.9               | 0.419         | 0             | 0.581        | 1.27        | 0.562   | 0.617   | --     | --     |
> | 1.0               | 0.911         | 0             | 0.089        | 0.21        | 0.086   | 0.109   | --     | --     |
> | 1.1               | 1.000         | 0             | 0            | 0.00        | 0       | 0       | --     | --     |

---

> > ### Author Response · Authors · 2025-12-04
> > **Answer to Reviewer rFyU [part 2]**
> >
> > **Table 2. Babilong QA2 early stopping experiments**
> >
> > | Q threshold       | stopped early | stopped later | perfect stop | Episode len | Fact EM | Fact F1 | Ans EM | Ans F1 |
> > |-------------------|--------------:|--------------:|-------------:|------------:|--------:|--------:|-------:|-------:|
> > | 0.00              | 0.000         | 0.994         | 0.006        | 6.00        | 0.996   | 0.499   | 0.884  | 0.884  |
> > | 0.10              | 0.000         | 0.498         | 0.502        | 2.86        | 0.996   | 0.845   | 0.944  | 0.944  |
> > | 0.20              | 0.000         | 0.036         | 0.964        | 2.29        | 0.996   | 0.949   | 0.976  | 0.976  |
> > | 0.30              | 0.006         | 0.010         | 0.984        | 2.25        | 0.990   | 0.952   | 0.970  | 0.970  |
> > | 0.40              | 0.008         | 0.002         | 0.990        | 2.24        | 0.988   | 0.953   | 0.970  | 0.970  |
> > | 0.50              | 0.008         | 0.000         | 0.992        | 2.23        | 0.988   | 0.954   | 0.972  | 0.972  |
> > | 0.60              | 0.016         | 0.000         | 0.984        | 2.21        | 0.980   | 0.948   | 0.968  | 0.968  |
> > | 0.70              | 0.042         | 0.000         | 0.958        | 2.16        | 0.954   | 0.934   | 0.948  | 0.948  |
> > | 0.80              | 0.112         | 0.000         | 0.888        | 2.06        | 0.884   | 0.905   | 0.884  | 0.884  |
> > | 0.90              | 0.177         | 0.000         | 0.823        | 1.92        | 0.820   | 0.861   | 0.830  | 0.830  |
> > | 1.00              | 0.930         | 0.000         | 0.070        | 0.22        | 0.070   | 0.107   | 0.230  | 0.230  |
> > | 1.10              | 1.000         | 0.000         | 0.000        | 0.00        | 0.000   | 0.000   | 0.000  | 0.000  |
> >
> >
> > Fact EM, Fact F1, Answer EM and Answer F1 measure the quality of the Q-RAG agent and the end-to-end QA system. In particular, Fact EM is the fraction of trajectories where all supporting facts are found, while Fact F1 also reflects the proportion of noisy facts among the retrieved ones. The columns “stopped_early”, “perfect_stop” and “stopped_later” evaluate the stopping criterion based on the Q-value. They are computed only over successful trajectories, where the ideal stopping time $t_{\text{oracle}}$ is the step when the last supporting fact is retrieved. Each of these three columns reports the fraction of cases where the Q-threshold criterion fires earlier than $t_{\text{oracle}}$, exactly at $t_{\text{oracle}}$, or after $t_{\text{oracle}}$ respectively. On the Babilong task, using the automatic stopping criterion yields almost perfect results for $q_{\text{threshold}}$ between 0.3 and 0.6. On HotpotQA, for $q_{\text{threshold}}$ values between 0.1 and 0.3 performance with this stopping criterion matches the performance achieved with fixed $T$ (see Table 2 in the paper).
> >
> > **[W2] Efficiency of Q-RAG**
> >
> > We agree that our efficiency claims require more substantial evidence. We strengthen this in the revised manuscript with a dedicated runtime and complexity analysis (Appendix E, Method Complexity and Efficiency). In this section, we analyze the asymptotic time and space complexity of Q-RAG as a function of the context length, and we add training curves on our datasets that show Q-RAG performance saturates within the first 12 hours of training. A key advantage in training efficiency is that Q-RAG fine-tunes only the embedder (~0.5B parameters) via reinforcement learning, avoiding the computational expense of fine-tuning a full, much larger LLM (for example 7B+ parameters). The following table compares the resource requirements and training time of Q-RAG against recent multi-step retrieval methods:
> >
> > | Method                  | Tunable Components | # Tunable Params | GPU Hours (Reported)        | Hardware Requirement |
> > | :---------------------- | :----------------- | :--------------- | :-------------------------- | :------------------- |
> > | Q-RAG (Ours)            | Embedder           | ~0.5B            | ~12                         | 1x A100-80GB         |
> > | BeamRetriever           | Reranker           | ~0.5B            | ~48                         | 1x A100-80GB         |
> > | Search-R1 / R1-Searcher | LLM                | 7B+              | ~hundreds (est. from paper) | 8x A100              |
> > | RAG-RL                  | LLM                | 7B               | ~500                        | 8x H100              |
> >
> > This comparison shows that Q-RAG achieves competitive performance while requiring orders of magnitude less compute and significantly cheaper hardware, making it accessible to a broader research community.

---

> > > ### Author Response · Authors · 2025-12-04
> > > **Answer to Reviewer rFyU [part 3]**
> > >
> > > **[W3] More details on experimental setup.**
> > >
> > > We added new section "Evaluation Details" and expanded "Training Details" section in the revisited version to provide enough information about our experimental setup.
> > >
> > > **1. Embedder Models and Architecture:**
> > >
> > > * For Open-Domain QA (HotPotQA, Musique) we used `multilingual-e5-large` as the foundation for both the state embedder \(E_s\) and the action embedder \(E_a\). This model was chosen for its strong general-purpose retrieval capabilities.
> > > * For Long-Context Benchmarks (Babilong, RULER) we used `facebook/contriever` as the base model. We initially started with this model, but in our experiments on a subset of tasks, the results obtained with `facebook/contriever` are fully reproducible when using `multilingual-e5-large` as well, so we did not find it necessary to recompute all completed experiments with the stronger `multilingual-e5-large`.
> > >
> > > In both cases, the state embedder \( E_s \) and action embedder \( E_a \) were initialized from the same base model but were fine-tuned as separate networks with independent parameters \( \theta_1 \) and \( \theta_2 \), as described in our method.
> > >
> > >
> > > **2. Reader LLMs for Baselines**
> > >
> > > To ensure a fair comparison, we standardized the reader LLM wherever possible. For Q-RAG we use Qwen3-4B as the reader for Needle-in-a-Haystack and temporal reasoning benchmarks, and QwQ-32B for open-domain QA tasks. For all reproduced baselines that use frozen LLMs, we used the same reader LLM as Q-RAG on the corresponding tasks (Beam Retriever, Graph Reader, Ablation baselines). Details on the specific LLMs used for each baseline are provided in Appendix sections "Training Details" and "Evaluation Details".
> > >
> > >
> > > **3. Chunking strategy**
> > >
> > > We split documents into fixed-length, non-overlapping chunks, aiming not to break sentences across chunk boundaries. The chunk length is primarily determined by the context window of the embedders used in our main experiments (512 tokens) and the number of retrieval steps. For Needle-in-a-Haystack and BabiLong we use a chunk length of 64 tokens. For open-domain QA tasks we set the chunk length as a function of the number of retrieval steps. In additional experiments with a `Alibaba-NLP/gte-multilingual-base` (8k context length) we use a chunk length of 256 tokens.

---

### Author Response · Authors · 2025-12-03
**Summary of reviewers feedback and authors response**

We thank all reviewers for their constructive critiques and thoughtful engagement with our work. We are grateful that the reviewers identified several consistent strengths in our submission, which affirm the value of our research. The comments have been invaluable in helping us improve our manuscript.

**Strengths:**
Reviewers uniformly highlighted the **novelty and technical soundness** of formulating multi-step retrieval as a reinforcement learning problem, coupled with the **exceptional resource efficiency** of training only the embedder rather than a full LLM. They also recognized the method's **strong empirical performance**, achieving state-of-the-art or highly competitive results across a diverse set of challenging long-context and multi-hop QA benchmarks, and noted the **rigorous ablation study** that validates our key design choices.

In response to common concerns, we have significantly strengthened the manuscript. We clarified the retrieval step budget and introduced a principled, Q-value-based early-stopping policy, with new experimental validation. To ensure reproducibility, we added comprehensive details on the embedder models, reader LLMs, and chunking strategy used, and some other configurations. Our claims of training efficiency are now substantiated with a direct comparison showing Q-RAG requires only a fraction of the GPU resources of recent baselines. Finally, we acknowledged the current reliance on supervised rewards and outlined a cautionary path toward using LLM-based feedback in future work.

In the revisited version of the paper, we added the following new sections that address common questions and weaknesses from reviewers:
* Appendix B: Early Stopping Experiments
* Appendix C: Sensitivity to Retrieval Budget
* Appendix E: Method Complexity and Efficiency
* Appendix H: Evaluation Details

Through these revisions, which encompass new analyses, detailed explanations, and a clarified scope, we have addressed the key points raised. We believe the revised paper now presents a compelling, well-supported, and accessible advancement in efficient multi-step retrieval for long-context RAG.

---

### Meta-Review · Area_Chair_8LzG · 2025-12-25

**Summary:**

This paper improves RAG by introducing multiple steps of retrieval rather than just a single step. Coupled with reinforcement learning, the authors show good results on long-context benchmarks. The reviewers are quite content with the paper, and except for one reviewer, there is a consensus the paper is ready for publication.

**Reviewer Concerns:**

presentation, narrow empirical validation

**Reviewer Scores:**

I can't tell

---

### Decision · Program_Chairs · 2026-01-26

Accept (Oral)